# GradMem: Learning to Write Context into Memory with Test-Time Gradient Descent

**Yuri Kuratov** [1 2]   **Matvey Kairov** [2]   **Aydar Bulatov** [1 2]   **Ivan Rodkin** [3 2]   **Mikhail Burtsev** [4]

## Abstract

Many large language model applications require conditioning on long contexts. Transformers typically support this by storing a large per-layer KV-cache of past activations, which incurs substantial memory overhead. A desirable alternative is *compressive memory*: read a context once, store it in a compact state, and answer many queries from that state. We study this in a context removal setting, where the model must generate an answer without access to the original context at inference time. We introduce **GradMem**, which writes context into memory via *per-sample test-time optimization*. Given a context, GradMem performs a few steps of gradient descent on a small set of prefix *memory tokens* while keeping model weights frozen. GradMem explicitly optimizes a model-level self-supervised context reconstruction loss, resulting in a loss-driven write operation with iterative error correction, unlike forward-only methods. On associative key–value retrieval, GradMem outperforms forward-only memory writers with the same memory size, and additional gradient steps scale capacity much more effectively than repeated forward writes. We further show that GradMem transfers beyond synthetic benchmarks: with pretrained language models, it attains competitive results on natural language tasks including bAbI and SQuAD variants, relying only on information encoded in memory.

## 1. Introduction

Large language models are increasingly deployed in settings where task-relevant information resides in long, external contexts: documents, codebases, tool interactions in

[1]AXXX, Cognitive AI Systems Lab, Moscow, Russia [2]MIRAI, Moscow, Russia [3]MBZUAI, Abu Dhabi, UAE [4]London Institute for Mathematical Sciences, London, UK. Correspondence to: Yuri Kuratov <yurakuratov@gmail.com>.

*Proceedings of the $43^{rd}$ International Conference on Machine Learning*, Seoul, South Korea. PMLR 306, 2026. Copyright 2026 by the author(s).

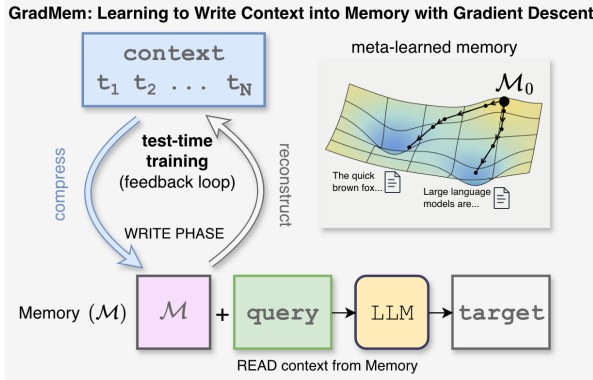

*Figure 1.* **GradMem learns to write context into memory via per-sample test-time optimization.** Given a context, GradMem performs a few test-time gradient updates on memory state to minimize a self-supervised reconstruction loss (WRITE). The memory initialization is meta-learned so that useful context representations can be written with only a few gradient steps. At inference, the model answers queries using only the optimized memory and the query (READ), without access to the original context.

agent workflows, and dialogue histories spanning multiple sessions (Lewis et al., 2020; Zhang et al., 2023; Gemini Team et al., 2024; Kimi Team, 2025). In these regimes, the challenge is not only to support long contexts, but to do so efficiently and *reusably*—ideally, the model reads a context once, stores what matters, and answers many queries without repeatedly re-processing the same tokens. The dominant approach is to retain intermediate activations via the KV-cache (and various compression schemes thereof), which reduces recomputation but can impose substantial memory overhead and does not naturally produce a portable representation of the context. A complementary alternative is to provide the model with a *compact memory state* that is constructed from a context and then reused across subsequent queries. Crucially, many applications require incorporating new information *without retraining or fine-tuning the full model*: we want to adapt the model to the current context by writing into a separate memory representation, while keeping the pretrained parameters fixed.

Recent work on *test-time training* shows that a model can adapt to the current context via gradient-based updates dur-

ing inference, and that iterative optimization of input embeddings can losslessly encode thousands of tokens given enough steps (Sun et al., 2025; Kuratov et al., 2025). Motivated by this observation, we introduce **GradMem**.[1] GradMem writes context into memory by *direct per-sample optimization* at test time (Figure 1). Specifically, GradMem treats embeddings of special memory tokens as *writable state* and performs a small number of *gradient descent* updates on this state for each context. This is *test-time training in the literal sense*: during inference, we execute a short inner-loop optimization on the current example. Crucially, GradMem cleanly separates *memory* from *model weights*: the base model parameters remain fixed, while adaptation to new contexts occurs solely through updates to the memory state. Unlike forward-only writing rules, this loss-driven inner loop provides per-example feedback, enabling GradMem to iteratively correct write errors as it forms a compact memory representation.

A key design choice in GradMem is the use of an *explicit, model-level WRITE objective* that is independent of the downstream supervision. In this paper, we focus on a simple self-supervised WRITE objective—*reconstruction*—computed from the language model's own predictions and backpropagated to the memory tokens. Because the objective is explicit, GradMem provides a direct way to trade compute for compression: additional gradient steps lead to a better memory state.

The intuition behind GradMem is simple. First, standard training with SGD can be viewed as a mechanism that *writes data into parameters* of a model via gradient updates (i.e., train set memorization); analogously, we treat memory as a parameter-like state to store the current context. Second, unlike one-shot forward writing (e.g., with text encoders), optimization provides an explicit signal of *what has not been encoded yet*: the reconstruction loss concentrates on the parts of the context that the model currently predicts poorly. Thus, gradient-based writing naturally prioritizes novel, unpredictable or high-entropy inputs and iteratively reduces reconstruction error. Third, while *lossless* context encoding via iterative optimization is known to be possible, it typically requires *hundreds to thousands* of gradient steps to achieve near-perfect reconstruction (Kuratov et al., 2025). In contrast, GradMem targets the few-step regime: by meta-learning the memory initialization and model parameters, we enable effective context writing with only a small number of test-time gradient steps.

We evaluate GradMem primarily on associative KV-retrieval task under context removal setting, a clean synthetic benchmark that directly measures how much information can be stored in a fixed-size memory. Across a wide range of set-

tings, GradMem stores more key–value pairs than forward-only methods that encode the context into memory with the same memory size. Our results also show that *how* the memory state is updated matters as much as *how many times* it is updated: even a single gradient-based WRITE update can write more information than a single forward-only update, and additional gradient descent steps further increase capacity. In contrast, repeating WRITE using only forward operations (e.g., re-processing the context multiple times) yields much weaker or less consistent gains. Beyond this synthetic setting, we study how performance varies with the number of WRITE steps, context length, and demonstrate that the same task-agnostic reconstruction objective transfers to pretrained language models on natural language tasks such as QA on bAbI, short SQuAD variants, and language modeling.

This paper makes the following **contributions**:

1. **GradMem: gradient-based context memorization.** We introduce GradMem, a memory mechanism that encodes a context into a compact memory state by performing a small number of test-time gradient descent steps on memory tokens while keeping the base model weights fixed. GradMem constructs memory using an explicit self-supervised WRITE objective (context reconstruction) computed at the model level, without requiring specialized per-layer memory update rules.

2. **Few-step gradient writing.** We show that a small set of memory tokens can be meta-trained so that $K \leq 5$ gradient descent steps reliably write task-relevant information into memory, enabling downstream tasks prediction with the original context removed.

3. **Gradient-based memory updates outperform forward-only writing.** On associative retrieval, gradient-based updates store substantially more information in a fixed-size memory state than WRITE mechanisms that use only forward computation. Moreover, increasing the number of gradient updates consistently improves memory capacity, whereas repeating forward-only writes provides limited or inconsistent gains.

4. **Capacity scaling and transfer to natural language.** We characterize how performance scales with the number of WRITE steps and context length on associative retrieval, and provide evidence of transfer to pretrained language models on natural language tasks (e.g., bAbI, SQuAD variants, language modeling) using the same task-agnostic reconstruction objective.

---

[1]Code is available at https://github.com/yurakuratov/gradmem.

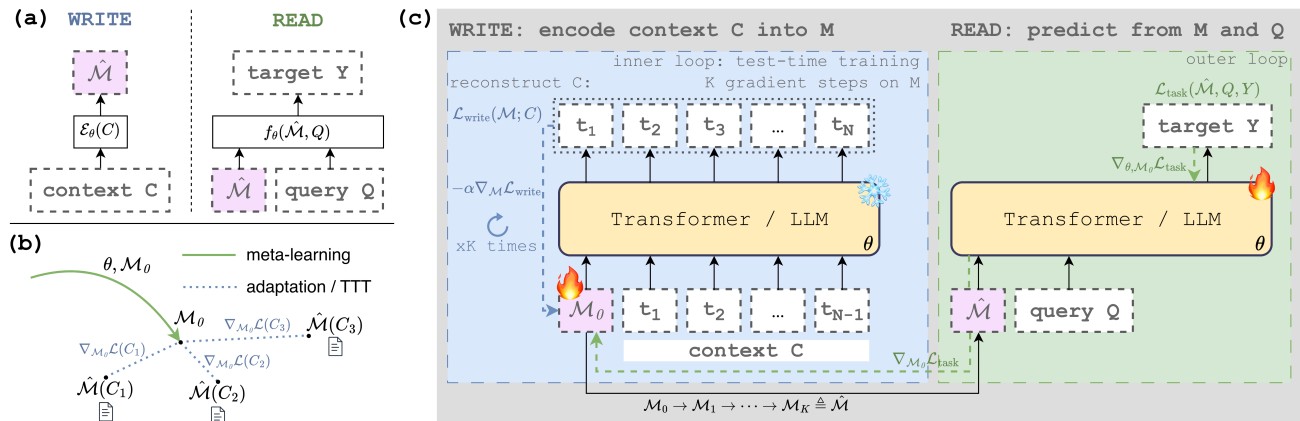

*Figure 2.* **GradMem overview. (a)** Each task sample is represented as context $C$, query $Q$, and target $Y$. A context encoder $E_\theta$ compresses $C$ into a fixed-size memory $\hat{\mathcal{M}}$ in a WRITE phase, and the model predicts $Y$ from $[\hat{\mathcal{M}}; Q]$ in a READ phase, without access to $C$. **(b)** Meta-learning view: a shared initialization $\mathcal{M}_0$ and model parameters $\theta$ are learned across training examples (outer loop), while at test time each context $C_i$ adapts its own memory $\hat{\mathcal{M}}(C_i)$ via a few gradient steps (dotted trajectory). **(c)** Test-time gradient descent on memory. Starting from the meta-learned initialization $\mathcal{M}_0$, GradMem updates the per-sample memory state during WRITE with $K$ steps of gradient descent on the context reconstruction loss $\mathcal{L}_{\text{write}}(\mathcal{M}; C)$. At READ, the model predicts the task target using only $[\hat{\mathcal{M}}; Q]$. During training, $\theta$ and $\mathcal{M}_0$ are optimized by backpropagating through the WRITE inner loop, so the model learns to use gradient descent as an operation that writes useful information about $C$ into memory; at inference, only the per-sample memory state is updated.

## 2. GradMem

### 2.1. Problem Setup: Context Removal Setting

Many sequence modeling problems can be expressed by separating (i) external information that can be used, (ii) a task specification, and (iii) the desired output. We formalize this by representing each task instance as three sequences: *context $C$*, *query $Q$*, and *target $Y$*. Our goal is to enable prediction of $Y$ from $Q$ without direct access to $C$ at inference time, by first compressing $C$ into a small, fixed-size memory $\mathcal{M}$.

The context $C$ contains information that the model can use, but which may be long or expensive to repeatedly process (e.g., a document, a list of facts, a repository codebase, or previous dialogue). The query $Q$ specifies what should be done with this information (e.g., a question, a key for retrieval, an instruction, or a prompt). The target $Y$ is the sequence to be predicted.

Let $f_\theta$ be a causal language model parameterized by $\theta$. We use $f_\theta(Y \mid X)$ to denote the probability assigned by the model to an output sequence $Y$ conditioned on an input sequence $X$ under the standard autoregressive factorization. In the standard causal language modeling setting, the model conditions on the concatenation of context and query:

$$f_\theta(Y \mid C, Q) \triangleq f_\theta\big(Y \mid [C; Q]\big). \tag{1}$$

This approach requires repeatedly attending to the full context for each query at increased compute cost. We instead consider a memory-augmented view with a **WRITE/READ** phase decomposition. We introduce a memory representa-

tion $\mathcal{M}$ (e.g., KV-cache, or $m$ input vectors of dimension $d$, or recurrent state) and define two phases:

**WRITE (encode context into memory).** A *context encoder* produces a memory state from the context:

$$\mathcal{M} = \mathcal{E}_\theta(C). \tag{2}$$

**READ (decode using memory and query).** The model predicts the target from the memory and the query:

$$f_\theta(Y \mid \mathcal{M}, Q) \triangleq f_\theta\big(Y \mid [\mathcal{M}; Q]\big). \tag{3}$$

The central evaluation constraint we consider is the *context removal*: during READ phase, the model does not have direct access to the original context $C$. All information needed to predict $Y$ must pass through the memory state $\mathcal{M}$ computed in WRITE phase. Under this setting (Figure 2a), a method is considered successful if the memory $\mathcal{M}$ captures enough task-relevant information from $C$ to solve the task *using memory and query only*.

### 2.2. GradMem: Test-Time Gradient Descent Memory

**GradMem** *directly optimizes* the memory representation $\mathcal{M}$: for every example, it performs *test-time training* by running a few gradient descent steps. Crucially, parameters of the model are frozen; instead, only the memory states $\mathcal{M}$ are trained on the current context $C$, resulting in a context-relevant representation for the subsequent READ phase.

**Memory parameterization.** We represent memory as $m$ vectors of dimension $d$, $\mathcal{M} \in \mathbb{R}^{m \times d}$. In a decoder-only

transformer, these vectors are used as *prefix embeddings* prepended to the model input. GradMem maintains a meta-learned initialization $\mathcal{M}_0$: a shared learned starting memory state used for all examples. At test time, $\mathcal{M}_0$ and model parameters are fixed; each context initializes from $\mathcal{M}_0$ and updates only its per-example memory for $K$ WRITE steps, producing $\mathcal{M}_K$. Whereas a transformer stores a KV-cache of size num_layers $\times N \times d \times 2$ for a context of length $N$, GradMem memory $\mathcal{M}$ is independent of context length and requires only $m$ $d$-dimensional vectors.

GradMem is closely related to Test-Time Training (TTT) layers (Sun et al., 2025), where an update is performed by gradient descent *online per token* (or small token mini-batches). The self-supervised objective in TTT is typically an $\ell_2$ reconstruction loss on the *layer input $x_i$*. TTT layers reconstruct layer inputs/activations, while GradMem reconstructs the context tokens, and does so once per context rather than at every layer and every token. Conceptually, instead of maintaining and updating a separate adaptive state in every layer, GradMem concentrates all test-time adaptation into a single memory state at the model input. Appendix J ablates this input-level memory parameterization against direct per-layer trainable KV-cache memory. Appendix A provides a broader comparison to long-context models, context-compression methods, fast-weight memories, and test-time-training works.

**WRITE: optimize memory to encode the context.** Given a context sequence $C = (t_1, \ldots, t_N)$ of $N$ tokens, GradMem uses an explicit WRITE objective that is task-agnostic and depends only on the ability of the model to reconstruct the context when conditioned on memory:

$$\mathcal{L}_{\text{write}}(\mathcal{M}; C) = -\sum_{i=1}^{N} \log f_\theta(t_i \mid [\mathcal{M}; t_{<i}]), \quad (4)$$

i.e., an autoregressive cross-entropy loss over the context tokens computed while prepending the current memory $\mathcal{M}$. Intuitively, minimizing $\mathcal{L}_{\text{write}}$ forces memory $\mathcal{M}$ to encode information about $C$ that is *not* predictable from the prefix $t_{<i}$ alone (e.g., in high-entropy, novel or surprising contexts). In this setting, reducing the $\mathcal{L}_{\text{write}}$ loss requires the model to use the fixed-size prefix $\mathcal{M}$ for storing context content.

Starting from the meta-learned initialization $\mathcal{M}_0$, GradMem performs $K$ steps of gradient descent *on the memory parameters only*:

$$\mathcal{M}_{k+1} = \mathcal{M}_k - \alpha \nabla_{\mathcal{M}_k} \mathcal{L}_{\text{write}}(\mathcal{M}_k; C), \quad (5)$$

where $\alpha$ is a WRITE-phase learning rate. We denote the final memory by $\hat{\mathcal{M}} \triangleq \mathcal{M}_K$, and define the context encoder as the composition of these optimization steps:

$$\hat{\mathcal{M}} = \mathcal{E}_\theta(C) \triangleq \text{GD}_K(\mathcal{M}_0, \mathcal{L}_{\text{write}}(\cdot; C)). \quad (6)$$

In practice, the update in Equation (5) can be stabilized with standard techniques such as gradient clipping. We also augmented it with (i) a learned linear layer applied to the memory before/after the updates and (ii) separate prediction heads for the WRITE and READ phases (we discuss these implementation variants in Appendix B).

**READ: predict only from memory and query.** In the READ phase, the model receives only $\hat{\mathcal{M}}$ and the query $Q$ and predicts the target:

$$f_\theta(Y \mid \hat{\mathcal{M}}, Q). \quad (7)$$

The overall training objective is the downstream task loss (e.g., next-token cross-entropy on $Y$) computed in the READ phase under context removal:

$$\mathcal{L}_{\text{task}}(\hat{\mathcal{M}}, Q, Y) = -\log f_\theta\Big(Y \mid \hat{\mathcal{M}}, Q\Big). \quad (8)$$

During training, we minimize $\mathcal{L}_{\text{task}}(\hat{\mathcal{M}}, Q, Y)$ w.r.t. $\theta$ and $\mathcal{M}_0$ by differentiating through the WRITE phase optimization steps that produce $\hat{\mathcal{M}}$. In this way, the model learns to use few gradient descent optimization steps as an operation to write useful information about current context $C$ into memory. Importantly, the WRITE objective $\mathcal{L}_{\text{write}}$ is not designed for any specific downstream task; it is a generic reconstruction loss used to form a memory state. GradMem training is summarized in Figure 2c.

GradMem can be viewed through a meta-learning lens (Figure 2b, Finn et al. (2017)): the WRITE phase performs a small number of per-sample optimization steps on $\mathcal{M}$, while the model parameters $\theta$ and the shared initialization $\mathcal{M}_0$ are trained so that these few steps reliably produce useful memories. In this view, the WRITE updates in Equation (5) correspond to an *inner optimization (inner loop)* over per-sample memory variables, and the task loss in Equation (8) defines an *outer objective (outer loop)* used to learn $\theta$ and $\mathcal{M}_0$ across training examples. We backpropagate through the WRITE optimization (yielding second-order gradients).

A common way to implement the WRITE phase is with a forward-only context encoder that maps $C \mapsto \mathcal{M}$ in a single pass (e.g., an encoder network, or a recurrent/segment-level state update) (Le & Mikolov, 2014; Kiros et al., 2015; Cer et al., 2018; Li et al., 2025; Gao et al., 2024; Rae et al., 2020; Chevalier et al., 2023; Behrouz et al., 2025c). Such encoders must learn to produce a useful memory *without any per-sample feedback* at inference time: once $\mathcal{M}$ is emitted, the write operation cannot verify whether the context was encoded well enough, nor correct mistakes made during compression. GradMem instead treats memory formation as an explicit optimization problem. By defining a task-agnostic reconstruction objective $\mathcal{L}_{\text{write}}(\mathcal{M}; C)$ and taking a small number of gradient descent steps on $\mathcal{M}$, GradMem obtains a direct signal of *how well* the current memory

explains the context and can iteratively refine $\mathcal{M}$ to reduce this loss. This iterative, loss-driven write mechanism is more expressive than a fixed forward computation: it can allocate compute to the specific context at hand, correct earlier write errors, and trade additional test-time compute (more gradient steps) for improved memory capacity.

# 3. Experiments and Results

## 3.1. Datasets

All experiments follow the context removal setting (Section 2.1) with two input segments for each of READ and WRITE phases. Each example is decomposed into a *context* $C$, a *query* $Q$, and a *target* $Y$.

**Associative KV-retrieval.** Associative retrieval is our main synthetic and controllable benchmark for comparing different memory mechanisms. Each example contains $N$ key–value pairs $(k_i, v_i)$, where each key and each value consists of 2 symbols from a 62-character vocabulary. The context is a sequence of key–value pairs with special delimiters:

$$C = {!}\,k_1{:}v_1\,{!}\,{!}\,k_2{:}v_2\,{!}\cdots{!}\,k_N{:}v_N\,{!}$$

The query $Q$ asks for the value associated with key $k_j$ and the target $Y$ is the corresponding value:

$$Q = {?}\,{!}\,k_j{:},\qquad Y = v_j.$$

The model can answer correctly only if the mapping from keys to values is written into memory during WRITE phase.

**bAbI** (Weston et al., 2016) is a question answering benchmark that tests reasoning over stories (e.g., tracking entities, locations, and interactions across multiple sentences). We use tasks QA1–QA5, which progressively increase the amount of multi-sentence composition required: QA1–QA3 require combining one, two, or three supporting facts, while QA4–QA5 require reasoning over two-argument and three-argument relations expressed across sentences. Each example consists of a story (a sequence of sentences) followed by a question and a short answer. We define the context $C$ as the story text, the query $Q$ as the question, and the target $Y$ as the ground truth answer string.

**SQuAD** (Rajpurkar et al., 2016) is an extractive question answering dataset, where each example consists of a paragraph, a question, and an answer span within the paragraph. We construct a short context variant (Short SQuAD) to control context length and isolate whether GradMem's writing mechanism transfers to natural language by extracting sentences containing the annotated answer span only. We define the context $C$ as the passage with the answer span, the query $Q$ as the question, and the target $Y$ as the answer text.

**Language Modeling** task evaluates next-token prediction ability of the model, where conditioning on a preceding context typically reduces the cross-entropy loss (perplexity) on subsequent tokens. We use WikiText-103 (Merity et al., 2017) (`wikitext-103-raw-v1`) and form examples by taking contiguous 256-token chunks. For segmented models (RMT, ARMT, GradMem), we split each chunk into two 128-token segments: the first segment is the context $C$ and the second is the target $Y$. There is no separate query in this setup ($Q = \emptyset$): after writing $C$ into memory, the model must predict the continuation $Y$ from memory alone. We report average cross-entropy on the last 128 tokens of each segment (segment 2, target). For non-segmented models, we compute the same metric by averaging token-level losses over positions 128–255 of the chunk, enabling a position-matched comparison to the segmented setting.

## 3.2. Baselines

**Full-Attention Transformer** This trivial baseline presents an upper bound of what can be memorized: the memory of a Transformer is uncompressed and contains all input hidden states. For associative retrieval experiments we train a small 4-layer Llama model (Touvron et al., 2023), and for downstream tasks we finetune pretrained GPT-2 (Radford et al., 2019) and Pythia (Biderman et al., 2023) models.

**Mamba** We include pretrained Mamba-2 model (130M) as a strong state-space baseline for sequence modeling (Dao & Gu, 2024). Mamba replaces quadratic self-attention with a selective state-space model (SSM) update, yielding linear-time processing in sequence length while retaining strong performance via input-dependent selection/gating. In our experiments, Mamba provides a natural comparison point: it maintains an internal recurrent state that summarizes the prefix and can be reused when processing subsequent tokens, without requiring an explicit attention cache.

**RMT** The Recurrent Memory Transformer (RMT) (Bulatov et al., 2022) is used as the straightforward forward-only memory write baseline. RMT splits the context into segments and iteratively processes them one after another with an LLM. It passes special memory vectors alongside segment tokens in order to memorize important information and reuse it. We use 2-segment version of RMT: the first segment contains the context, and the second one starts with the query and generates the answer. In this setting RMT is fully equivalent to GradMem except for the memory write operation that is performed by the forward pass. For associative retrieval experiments we wrap the 4-layer llama model with hidden_size 128 and 4 attention heads, while for our natural language experiments we wrap the GPT-2 (124M) model. The same applies to the ARMT model. For more training details see Appendix B.

**ARMT** Associative Recurrent Memory Transformer (Rodkin et al., 2024) accumulates information segment by segment into a small set of memory tokens and stores them in

*Table 1.* **KV-retrieval: gradient-based WRITE outperforms forward-only WRITE.** Exact match retrieval accuracy (mean±std over 3 runs) for predicting a 2-token value from a 2-token key as the number of key–value pairs increases. **Upper bound:** a standard Transformer with full KV-cache retains all past activations. **Per-layer memory baselines:** Mamba and ARMT maintain state in every layer. **Same memory state:** RMT (forward-only write) and GradMem both use the same base architecture and the same memory size of $m=8$ vectors, but a different WRITE rule. Repeating the forward-only WRITE by re-reading the same context segment (x2–x5) gives limited or inconsistent gains, whereas additional gradient-based WRITE steps (x2, x5) consistently improve retrieval at larger numbers of pairs.

| | NUMBER OF KV-PAIRS | | | | | |
|---|---|---|---|---|---|---|
| MODEL | 4 | 8 | 16 | 32 | 64 | 96 |
| TRANSFORMER: $\mathcal{M}$=KV-CACHE | $100.0_{\pm 0.0}$ | $100.0_{\pm 0.0}$ | $99.8_{\pm 0.0}$ | $99.8_{\pm 0.3}$ | $96.5_{\pm 2.9}$ | $98.8_{\pm 0.0}$ |
| MAMBA: $\mathcal{M}$=PER-LAYER RECURRENT STATE | $99.9_{\pm 0.1}$ | $98.9_{\pm 0.4}$ | $98.7_{\pm 0.2}$ | $90.2_{\pm 10.2}$ | $95.2_{\pm 0.1}$ | $\mathbf{92.2}_{\pm 0.4}$ |
| ARMT: $\mathcal{M}$=PER-LAYER ASSOCIATIVE MATRIX | $99.0_{\pm 0.3}$ | $98.5_{\pm 0.5}$ | $97.4_{\pm 0.3}$ | $54.9_{\pm 2.1}$ | $22.6_{\pm 3.9}$ | $15.2_{\pm 0.2}$ |
| FORWARD-ONLY WRITE (RMT): $\mathcal{M}$=8 MEM VECTORS | $\mathbf{100.0}_{\pm 0.0}$ | $\mathbf{100.0}_{\pm 0.0}$ | $45.5_{\pm 0.2}$ | $44.3_{\pm 0.0}$ | $19.3_{\pm 3.1}$ | $12.9_{\pm 0.2}$ |
| X2 MEMORY UPDATES | | | $69.6_{\pm 28.1}$ | $18.7_{\pm 3.4}$ | – | – |
| X3 MEMORY UPDATES | | | $60.0_{\pm 42.0}$ | $38.1_{\pm 0.0}$ | – | – |
| X4 MEMORY UPDATES | | | – | $31.5_{\pm 0.0}$ | – | – |
| X5 MEMORY UPDATES | | | – | $37.0_{\pm 1.7}$ | – | – |
| GRADMEM: $\mathcal{M}$=8 MEM VECTORS | $100.0_{\pm 0.0}$ | $99.7_{\pm 0.0}$ | $96.3_{\pm 0.9}$ | $86.9_{\pm 0.5}$ | $58.6_{\pm 0.7}$ | $32.6_{\pm 0.1}$ |
| X2 MEMORY UPDATES | $100.0_{\pm 0.0}$ | $100.0_{\pm 0.0}$ | $99.6_{\pm 0.1}$ | $98.3_{\pm 0.1}$ | $72.8_{\pm 0.5}$ | $34.2_{\pm 0.1}$ |
| X5 MEMORY UPDATES | $100.0_{\pm 0.0}$ | $100.0_{\pm 0.0}$ | $100.0_{\pm 0.0}$ | $\mathbf{99.9}_{\pm 0.1}$ | $\mathbf{99.1}_{\pm 0.3}$ | $88.4_{\pm 2.3}$ |
| X1, W/O 2ND-ORDER META-LEARNING | | $12.9_{\pm 8.1}$ | $3.0_{\pm 0.6}$ | – | – | – |
| X2, W/O 2ND-ORDER META-LEARNING | | $46.7_{\pm 8.3}$ | $4.2_{\pm 0.7}$ | – | – | – |

an associative matrix with a DeltaNet-style update (Yang et al., 2024). ARMT performs a forward-only WRITE: as it processes a segment, it produces memory tokens and writes them into the associative memory on each layer, which is then queried over future segments. In this paper we use a two-segment setup as for RMT. This makes ARMT a strong baseline for assessing whether *gradient-based* writing (GradMem) into memory tokens stores more task-relevant information than *forward-only* writing into per-layer associative matrices.

### 3.3. Results on KV-retrieval Task

We start with associative KV-retrieval and organize compared methods into four groups (Table 1). First, a standard Transformer that attends to the full context serves as an **upper bound**: it is not a compressive-memory method, since it retains all past activations (KV-cache). Second, sequence models with *per-layer* memory (e.g., Mamba and ARMT) implement a different memory interface: their state is distributed across layers and is therefore not directly matched to a single model-level memory state, their state sizes are much larger (see Table 6 for exact memory-state sizes and Appendix G for state-matched Mamba results). We include them as strong reference points, but focus our main comparison on methods that write into the same memory parameterization (RMT, GradMem). Third, we consider **forward-only writing** into a memory (RMT), which updates memory purely through forward computation, uses the same base architecture, the same model-level memory state, and the same memory size, differing only in the WRITE rule. Finally, we evaluate **GradMem**, which uses the same base architecture and the same memory size ($m=8$ memory vectors),

but differs in the WRITE rule: GradMem updates memory tokens by a small number of test-time gradient steps. All models we compare here have 4 layers and 128-dim (for transformer-based models we use 4 attention heads). Unless stated otherwise, we use the same number of KV-pairs and memory updates for training and inference (RMT, GradMem). Once a method's accuracy dropped sharply at smaller $N$, we did not scale it further (entries marked with "–" in the Table 1).

**Gradient-based WRITE improves performance and scales with more steps.** Figure 3 compares forward-only writing (RMT) with GradMem under the same memory size, and varying the number of WRITE updates. We can see three trends. (i) Gradient-based updates of a model-level memory state are effective: by directly optimizing memory tokens per example at test-time, GradMem writes context information into a compact memory and attains high retrieval accuracy compared to RMT with forward-only memory update. (ii) Even a single gradient-based WRITE step ($K=1$) substantially outperforms a forward-only write at the same memory size. (iii) Allocating more WRITE compute to gradient updates further improves performance: increasing to $K=5$ enables accurate retrieval at much larger numbers of key–value pairs.

Appendix F sweeps memory size and WRITE steps, showing that larger $K$ improves memory efficiency at fixed $m$, but larger contexts still require larger memory. Appendix H additionally compares GradMem with TTT-Linear (Sun et al., 2025) and LaCT (Zhang et al., 2026) on the same KV-retrieval setup.

**Repeated forward writes provide limited gains com-**

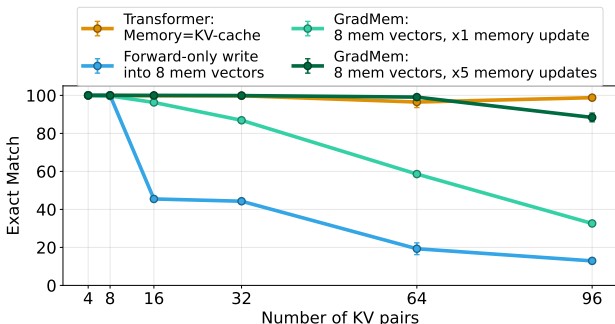

*Figure 3.* **Gradient-based memory updates (GradMem) outperform forward-only updates at the same memory size.** With a memory state of 8 vectors, GradMem retrieves any of 16 key–value pairs with 95% accuracy, whereas a forward-only update rule stores only 8 pairs with high accuracy. More gradient steps increase the capacity of the same 8-vector memory to 96 pairs at 88% retrieval accuracy. Transformer with KV-cache as non-compressive memory serves as an upper bound.

**pared to gradient updates.** Table 1 reports the full set of results on KV-retrieval. Among methods that write into a *single memory state* of 8 vectors, GradMem consistently achieves higher accuracy than forward-only writing (RMT). Moreover, repeating the forward-only WRITE by *re-reading the same context segment multiple times* yields weak or inconsistent improvements: each RMT pass replaces the previous memory with a new forward-computed memory, rather than accumulating or explicitly correcting it. By contrast, additional gradient-based WRITE steps use the reconstruction loss as per-example feedback and reliably increase performance. These results support the conclusion that gradient-based updates provide a more expressive write operation than forward-only computation, and additional gradient steps offer a way to trade test-time compute for better memory quality. We also found that the w/o 2nd-order meta-learning ablation (Table 1) does not reach strong performance. It still performs test-time memory updates and provides a first-order signal to the shared starting memory state $\mathcal{M}_0$, but omits the full second-order meta-gradient through the WRITE updates. This suggests that the second-order meta-learning signal is important for learning a strong gradient-based WRITE rule, so we use the full second-order variant (MAML, Finn et al. (2017)) in further experiments.

### 3.4. Scaling Inference Compute with More WRITE Iterations

In GradMem, we find that increasing the number of WRITE iterations at evaluation time provides a substantial accuracy lift on KV-retrieval task. Figure 4(a) shows a trend across settings with different numbers of key–value pairs where for some models extrapolating from the training-time value $K_{\text{train}}$ to larger $K_{\text{eval}}$ yields improvements in exact match. Importantly, these gains are obtained with *fixed* model parameters and therefore reflect the effect of allocat-

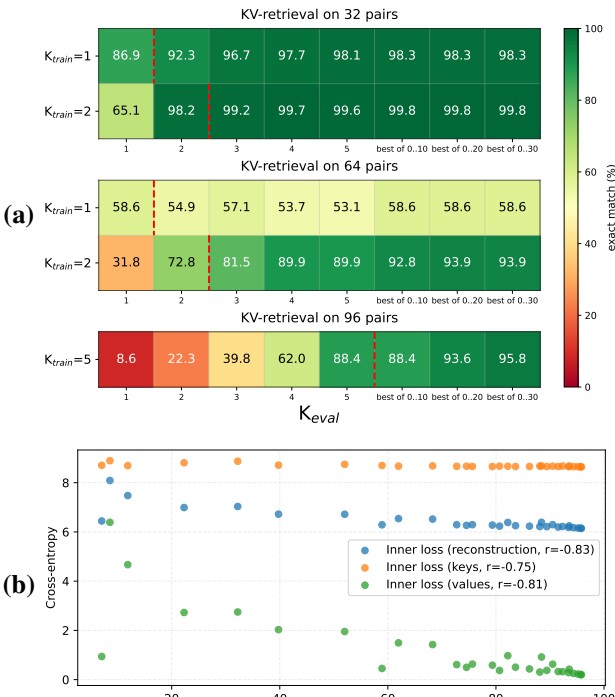

*Figure 4.* **More gradient steps at test-time lead to better performance without fine-tuning.** (a) Results for $K_{\text{train}}$ values of GradMem setups that could not achieve 99% exact match during fine-tuning. Red dashed lines denote the beginning of extrapolation. (b) Downstream task quality (Exact Match) correlates with WRITE objective in inner loop (reconstruction). Notably, GradMem reconstructs *values* better than *keys*, despite the WRITE objective treating them equally. Results on 96-pair KV-retrieval.

ing additional WRITE refinement steps at inference. We connect this effect to better convergence on the inner task. By decomposing the inner loss into loss on key or value tokens (Figure 4(b)) we observe that inner loss correlates strongly with exact match (more results are in Appendix E).

This behavior induces a practical compute–accuracy trade-off and a way to scale computation beyond training without re-optimization. Larger $K_{\text{train}}$ is expensive under our meta-learning objective, and it becomes increasingly cumbersome to fine-tune models that must backpropagate through long WRITE trajectories—particularly when longer contexts and larger $K$ are required. In contrast, increasing $K_{\text{eval}}$ only at inference time shifts this cost to evaluation, enabling higher accuracy without any fine-tuning while preserving a small-$K$ training pipeline.

### 3.5. GradMem with Pre-trained Models on NLP Tasks

We next evaluate GradMem with pretrained language models on natural language benchmarks to test whether the gradient-based WRITE method and reconstruction-based WRITE objective can produce useful memory states outside the controlled KV-retrieval setting.

*Table 2.* **GradMem remains competitive on downstream language tasks.** Results on bAbI (QA1–5), SQuAD (short) shown as Exact Match (EM) in %, and language modeling (cross-entropy). RMT is the matched memory state baseline; full-context Transformers are upper bounds, while Mamba/ARMT use different and larger memory states. We use 32 memory tokens for SQuAD and LM, and 8 on bAbI. We report mean±std over 3 runs.

| | bAbI (EM↑) | | | | | SQuAD (EM↑) | LM (CE↓) |
|---|---|---|---|---|---|---|---|
| | QA1 | QA2 | QA3 | QA4 | QA5 | short | 128–255 |
| Input length (tokens) | $\sim$40 | $\sim$100 | $\sim$300 | $\sim$20 | $\sim$20 | $\sim$40 | 256 |
| *Full context models (upper bound)* | | | | | | | |
| GPT-2-124m | $100.0_{\pm 0.0}$ | $100.0_{\pm 0.0}$ | $99.8_{\pm 0.1}$ | $100.0_{\pm 0.0}$ | $99.4_{\pm 0.1}$ | $64.2_{\pm 0.3}$ | $2.72_{\pm 0.00}$ |
|   Limit context to 128 tokens | | | | | | | $3.20_{\pm 0.00}$ |
| Pythia-160m | $100.0_{\pm 0.0}$ | $99.7_{\pm 0.1}$ | $95.5_{\pm 2.7}$ | $100.0_{\pm 0.0}$ | $99.0_{\pm 0.1}$ | $48.9_{\pm 0.4}$ | $2.84_{\pm 0.05}$ |
| *Recurrent models* | | | | | | | |
| Mamba-130m-hf | $100.0_{\pm 0.0}$ | $100.0_{\pm 0.0}$ | $96.7_{\pm 0.2}$ | $100.0_{\pm 0.0}$ | $99.7_{\pm 0.09}$ | $63.3_{\pm 0.2}$ | $2.69_{\pm 0.00}$ |
| RMT (GPT-2) | $100.0_{\pm 0.0}$ | $93.9_{\pm 0.1}$ | $87.9_{\pm 0.4}$ | $100.0_{\pm 0.0}$ | $93.9_{\pm 6.9}$ | $42.6_{\pm 0.3}$ | $2.91_{\pm 0.00}$ |
| ARMT (GPT-2) | $100.0_{\pm 0.0}$ | $93.8_{\pm 0.6}$ | $92.3_{\pm 0.8}$ | $100.0_{\pm 0.0}$ | $98.9_{\pm 0.1}$ | $39.0_{\pm 0.2}$ | $2.85_{\pm 0.00}$ |
| GradMem (GPT-2, $K=1$) | $100.0_{\pm 0.0}$ | $94.2_{\pm 0.4}$ | $80.0_{\pm 0.2}$ | $100.0_{\pm 0.0}$ | $99.2_{\pm 0.1}$ | $38.1_{\pm 0.1}$ | $2.92_{\pm 0.00}$ |
| GradMem (GPT-2, increased $K$) | $100.0_{\pm 0.0}$ | $93.9_{\pm 0.5}$ | $79.3_{\pm 0.2}$ | $100.0_{\pm 0.0}$ | $99.2_{\pm 0.1}$ | $54.9_{\pm 0.4}$ | $2.91_{\pm 0.01}$ |

**GradMem remains competitive on downstream language tasks, with task-dependent gains from larger $K$.** The NLP task evaluation spans across three task groups: bAbI reasoning benchmark, single-sentence SQuAD QA and the LM task on the Wikitext dataset (see Table 2). The bAbI evaluation exhibits significant variability of scores on different reasoning tasks. The context size of QA1, QA4 and QA5 generally does not exceed 40 tokens, and all evaluated methods solve them well, matching the upper bound Transformer performance. QA2 and QA3 require to memorize more facts and detect them among more noise, which poses a challenge to compressive models. Mamba performs best among recurrent models because its memory operations are already learned during extensive pretraining, unlike those of GradMem and recurrent Transformers. ARMT follows closely; its strong performance on QA2 and QA3 can be attributed to its larger memory-state size relative to other models in its class and to the additional parameters in its associative layers. GradMem matches or outperforms forward-only RMT on all QA tasks except QA3 with the highest information density.

To reduce the effect of long, distractor-heavy contexts on performance we evaluate text understanding on Short SQuAD, where the context is restricted to the answer-containing sentence; for dataset details refer to Section 3.1 and Appendix B. GradMem with larger $K$ outperforms forward-only RMT and achieves the best performance among recurrent Transformers and even the non-compressive Pythia model, while GPT-2 still remains the upper bound. On the Wikitext language modeling task, recurrent models are required to compress diverse information from context to correctly predict the next token of the current segment. All compressive Transformers learn to use memory and noticeably outperform GPT-2 with context size 128; full-context GPT-2 remains the upper bound. ARMT with more capacious memory takes the lead, followed by RMT and GradMem. The benefit of larger $K$ is task-dependent: it gives a large gain on Short SQuAD and a small improvement on language modeling, but does not improve bAbI in these runs.

## 4. Discussion and Conclusions

**Compute and memory overhead.** A core cost of GradMem is that it backpropagates through the WRITE inner loop and, during training, differentiates through the unrolled optimization steps. This increases both compute and GPU memory usage compared to forward-only writers and standard full-context training, since the computational graph must be retained across WRITE steps. In practice, this also constrains the choice of attention implementation: common high-performance kernels such as FlashAttention (Dao et al., 2022) and PyTorch SDPA are designed for efficient first-order backpropagation, but do not support the higher-order differentiation required by GradMem training (i.e., taking gradients through the WRITE updates). As a result, we implement a custom double-backward that is both *more memory-efficient* and *faster* than a naive eager implementation, described in Appendix C. More broadly, the meta-learning literature provides more computationally efficient approaches, including first-order/implicit methods such as iMAML (Rajeswaran et al., 2019) and Reptile (Nichol et al., 2018), which could further improve the efficiency of GradMem meta-learning training.

We also test GradMem beyond GPT-2 scale with pretrained Llama-3.2-1B and Llama-3.2-3B models on text reconstruction (Appendix I). With only 1–2 memory vectors and 1–2 WRITE steps, GradMem reconstructs short text spans (32 tokens) while keeping the READ model frozen.

**When test-time WRITE compute is worthwhile.** At inference time, the WRITE phase with $K$ gradient steps is more expensive than a single forward pass over the context. Nevertheless, in applications where the same context is reused across multiple queries, the additional WRITE compute can be amortized: after encoding a long context $C$ into a compact memory $\mathcal{M}$ once, subsequent READ computations attend only over $[\mathcal{M}; Q]$ rather than $[C; Q]$. When $|C| \gg |\mathcal{M}|$, this can reduce per-query computation and memory, making gradient-based writing a more compute-reasonable option. We analyze this in Appendix D.

**Write objectives beyond reconstruction.** We use a simple token-level reconstruction objective for $\mathcal{L}_{\text{write}}$, which is task-agnostic and easy to apply across domains. Despite its simplicity, it is already sufficient to yield strong gains on associative retrieval and to transfer to natural language tasks. At the same time, reconstruction is unlikely to be optimal for all downstream tasks. Related work on TTT layers (Sun et al., 2025) already explored learning self-supervised reconstruction objectives via input transformations, but considered relatively restricted (e.g., linear) transformations. A promising direction is to learn WRITE objectives that better preserve task-relevant information, while retaining the key constraint that WRITE remains self-supervised on the available context.

**Conclusions.** We introduced GradMem, a WRITE/READ memory mechanism where a model *writes* a context into a small set of memory tokens by running a few steps of test-time gradient descent while keeping model weights fixed. Across controlled KV-retrieval experiments, this direct per-sample optimization gives a stronger WRITE rule than forward-only updates under the same architecture and memory size. Increasing the number of gradient WRITE steps reliably improves retrieval as the number of stored key–value pairs grows, unlike repeating forward WRITE passes. We further showed that the same reconstruction-based WRITE objective can be applied to pretrained language models in natural language tasks (bAbI, short SQuAD, language modeling). Overall, our results establish gradient-based test-time optimization of a model-level memory state as a promising alternative to forward-only memory writers, and motivate future work on more efficient training, scaling to longer contexts and larger models, and better self-supervised WRITE objectives.

## Impact Statement

This paper presents work whose goal is to advance the field of Machine Learning. There are many potential societal consequences of our work, none which we feel must be specifically highlighted here.

## Acknowledgements

Y.K., M.K., A.B., and I.R.'s work was supported by the Ministry of Economic Development of the Russian Federation (Agreement No. 139-15-2025-013, dated June 20, 2025, IGK 000000C313925P4B0002).

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

## Appendix Contents

## A. Related Work

**Long-context modeling and efficient attention.** A large body of work aims to extend the effective context length of transformers by changing the architecture or attention mechanism, thereby reducing the need for an explicit external memory. Compressive Transformers (Rae et al., 2020) augment the standard recurrent memory with a compressed memory stream, retaining a lossy summary of distant past activations. Recurrent memory transformers such as RMT (Bulatov et al., 2022) and their extensions (Chevalier et al., 2023) segment long inputs into chunks and pass a trainable state between segments, enabling processing of sequences far beyond the native context window. Other approaches leverage associative memories and efficient attention mechanisms to scale to long contexts (Rodkin et al., 2024; Ramsauer et al., 2021).

**Context compression and reusable representations.** In NLP, compression has long been studied for constructing compact sentence- or document-level representations (Le & Mikolov, 2014; Kiros et al., 2015; Cer et al., 2018), often via autoencoding pipelines (Bowman et al., 2016; Miao et al., 2016). In the context of large language models, input compression is used both to reduce the quadratic cost of self-attention and to create persistent representations that can be stored or reused. For compressed data storage in LLMs, one can use dense continuous vectors or memory tokens (Burtsev et al., 2020), but also LoRA parameters (Hu et al., 2022), intermediate hidden states (Li et al., 2025), associative memories (Rodkin et al., 2024; Ramsauer et al., 2021), or the KV-cache directly (Chari et al., 2025; Karami et al., 2025). Some works focus on compressing only part of the input, such as prompts (Lester et al., 2021; Li & Liang, 2021) or in-context examples (Ge et al., 2024; Eyuboglu et al., 2025), while others iteratively compress the entire sequence by splitting it into smaller chunks and processing them sequentially (Rae et al., 2020; Bulatov et al., 2022; Chevalier et al., 2023; Prakash et al., 2025). Approaches such as task vectors (Ilharco et al., 2023) and Cartridges (Eyuboglu et al., 2025) compress information from multiple task samples into persistent vectors that can be reused to steer the model on those tasks, amortizing the cost of processing across many downstream queries.

A straightforward way to compress information into memory is to use the model itself as an encoder, e.g., via segment-level memory tokens in RMT-style architectures (Bulatov et al., 2022; Chevalier et al., 2023; Gao et al., 2024). ICAE (Ge et al., 2024) and SelfCP (Gao et al., 2024) train base LLMs to compress context using autoencoding-style objectives and other losses targeted at text understanding (Li et al., 2025).

**Fast weights, delta rules, and associative memory** offer another path to context-dependent memory. Early work used fast weights as high-level controllers (Schmidhuber, 1992) or as associative storage over past representations (Hinton & Plaut, 1987). Recent formulations adapt fast weights to modern architectures, often in the form of associative memories or gated recurrent states that implement a learned approximation to a delta rule. Associative-memory transformers (Rodkin et al., 2024) and modern Hopfield networks (Ramsauer et al., 2021) can be interpreted as storing and retrieving key–value pairs in a continuous memory, with the write operation implemented purely via forward computation. These mechanisms generally rely on *forward-only* update rules that are applied once per token or timestep, without an explicit per-example optimization objective that is iteratively minimized for a given context.

**Test-time training.** GradMem is closely related to test-time training (TTT) methods that perform gradient-based adaptation during inference. TTT layers (Sun et al., 2025) introduce lightweight, sequence-dependent state that is updated online

using self-supervised objectives, typically by reconstructing layer activations at each timestep. A related line of work uses optimization as a way to compress large amounts of data into a fixed number of weights or embeddings. Early formulations viewed fast weights as auxiliary parameters that are updated via gradient-based rules (Schmidhuber, 1992). More recent work proposes unsupervised objectives for compressing context via test-time optimization (Sun et al., 2025), and adapts these ideas to transformers by introducing additional memory modules, optimization-based memory update rules, routing mechanisms and end-to-end training objectives (Behrouz et al., 2025c;a;b; Tandon et al., 2025). Recent methods such as LaCT (Zhang et al., 2026), and ATLAS (Behrouz et al., 2025a) also adapt sequence-dependent state at inference time. These methods typically maintain layer-local fast states or neural-memory modules and update them online while processing tokens or chunks, often using local self-supervised objectives over hidden activations or layer inputs. The study of Kuratov et al. (2025) shows that, by optimizing a simple reconstruction objective with gradient descent, it is possible to achieve extremely high compression ratios (up to $\sim$1500$\times$) into a single vector; however, this requires up to tens of thousands of gradient updates and produces representations that are primarily useful for text reconstruction rather than downstream tasks.

**Comparison to our approach** Our work is distinguished from prior memory and TTT approaches along three main axes. First, GradMem uses a *single input-level memory state* that is written once per context, rather than per-layer or per-token states updated online. Second, the WRITE mechanism is *explicitly optimization-based*: memory tokens are treated as parameters and updated by gradient descent on a model-level reconstruction loss, rather than via a learned forward-only update rule. Third, we target the *few-step* regime, meta-training the base model and memory initialization so that a small number of gradient steps ($K \leq 5$) suffices for effective writing, in contrast to hundreds or thousands of iterations used in prior embedding-optimization work. Table 3 provides a detailed comparison to TTT layers, and Table 4 compares GradMem with a broader range of test-time-training methods. Altogether, GradMem can be described as a model-level, context-level, multi-step gradient-based WRITE mechanism with explicitly controllable memory size, trained with second-order meta-learning through the whole model and inner optimization loop. Because the writable state is simply $m$ memory tokens, the memory budget is explicit and directly controllable. Moreover, GradMem can perform multiple gradient-descent WRITE steps on the same context memory, exposing an iterative compute–quality trade-off that is not captured by one-shot forward writing or local online update rules.

*Table 3.* Comparison of GradMem to test-time training (TTT) layers. Both approaches perform learning at inference time via a self-supervised loss, but differ in *what* is adapted (layer parameters vs. memory tokens), *when* it is adapted (token-level online vs. context-level WRITE), and *what* the self-supervised signal reconstructs (layer inputs/activations vs. context tokens).

|  | **TTT layers** (Sun et al., 2025) | **GradMem (ours)** |
| --- | --- | --- |
| Usage pattern | Sequence-modeling layer: updates state online while processing tokens | Explicit two-phase WRITE/READ setting |
| Inner-loop input | Token $x_t$ (or token mini-batches) | Whole context segment $C$ (WRITE once per context) |
| Test-time parameters | Layer-specific parameters $W_t$ (updated from the layer's inputs/activations), *per layer* | Prefix memory tokens $\mathcal{M}_k \in \mathbb{R}^{m \times d}$ (single memory state), *per model* |
| Self-supervised loss | Activation/input reconstruction, e.g. $\ell(W; x_t) = \|f(\tilde{x}_t; W) - x_t\|_2^2$ (or learned multi-view projections) | Context reconstruction $\mathcal{L}_{\text{write}}(\mathcal{M}; C)$ |
| Outer-loop objective | Next-token prediction (LM training) | Downstream task loss with $C$ removed at READ |
| Outer-loop parameters | Model params $\theta$ + reconstruction task/view params | Model params $\theta$ + memory init $\mathcal{M}_0$ (optional: memory projections / control tokens) |

*Table 4.* GradMem and TTT methods. GradMem is characterized by explicit model-level memory update operation, performed once per context.

| Property | TTT layers (Sun et al., 2025) | Titans (Behrouz et al., 2025c) | Atlas (Behrouz et al., 2025a) | LaCT / TTT Done Right (Zhang et al., 2026) | GradMem (ours) |
|---|---|---|---|---|---|
| Memory level | Layer | Layer | Layer | Layer | Model |
| Memory update | Per-token/small-mini-batch | Per-token/chunk + momentum/decay | Per-token/chunk with sliding window updates | Per chunk | One WRITE per full context $C$ |
| Inner-loop input | Token $x_t$ / views of $x_t$ | Key–value proj. $(k_t, v_t)$ of token $x_t$ | Local window of keys and values | Chunk of keys and values | Full context $C$ |
| Test-time parameters | Weights $W_t$, per layer | MLP mem. + momentum, per layer | MLP mem. + momentum + gates, per layer | Weights $W_t$, per layer | Single memory prefix $\mathcal{M} \in \mathbb{R}^{m \times d}$ |
| WRITE objective | View reconst., $L_2$ | KV recall, $L_2$ | Window KV recall, $L_2$ | Chunk KV recall, dot product | Context reconst. $\mathcal{L}_{\text{write}}(\mathcal{M}; C)$ |
| Outer-loop objective | Task loss / next-token prediction | Task loss / next-token prediction | Task loss / next-token prediction | Task loss; next-token prediction | Task loss at READ, context removed |
| Added outer-loop param. | Reconstruction task/view params | MLP/proj. weights, decay, momentum | Polynomial kernel, Muon, decay | Muon, momentum | Memory init $\mathcal{M}_0$ |

# B. Implementation, Training, and Hyperparameter Details

Code is available at https://github.com/yurakuratov/gradmem.

Table 5 summarizes the memory configuration and training initialization used across tasks. Table 6 lists models parameters. We keep the memory embedding dimension fixed to $d_{\text{mem}} = 64$ in all ARMT experiments, and vary only the number of memory tokens depending on the task. Unless otherwise noted, "from pretrained model" means initializing the base LM weights from a standard pretrained checkpoint (e.g., GPT-2 (124M) / Pythia (160M) / Mamba (130M)) and then fine-tuning with the corresponding memory mechanism.

For associative retrieval (AR), we train models from scratch and use a curriculum over context lengths: training for a larger number of key–value pairs is initialized from the final checkpoint obtained at a smaller length. For GradMem, the curriculum starts from 32 key–value pairs, as we found that the model achieves perfect retrieval on smaller contexts even without curriculum training. Increasing the number of inner-loop WRITE steps can make training less stable, since each additional step deepens the meta-gradient computation. In practice, the most effective stabilization was curriculum learning: we start with shorter sequences and smaller $K$, then gradually increase the difficulty.

For bAbI, most models are fine-tuned from pretrained checkpoints; the exception is RMT, which we found to be sensitive to initialization and therefore train it starting from a GPT-2 (124M) checkpoint already fine-tuned on bAbI. For language modeling, all models are fine-tuned from pretrained checkpoints; additionally, we report a variant where GradMem is initialized from an RMT checkpoint trained on the same language modeling objective.

For GradMem, we tune the inner learning rate $\alpha \in [0.01, 10]$. We observe that model performance does not depend strongly on its value, as long as it is in a reasonable range. We found $\alpha = 0.4$ to be a relatively strong default value for experiments on NLP tasks. For the KV-retrieval task, we still perform small search over $\alpha$ and report the scores for the best setup with every $K$, but we do not tune $\alpha$ exhaustively in every experiment. We also augment both GradMem and RMT with (i) a learned linear layer applied to the memory before/after the updates and (ii) separate prediction heads (output embeddings) for the WRITE and READ phases. These augmentations are used in all experiments unless stated otherwise. We backpropagate through the unrolled WRITE optimization by default; we explicitly note experiments that disable this meta-learning path (e.g., Table 1, w/o 2nd-order meta-learning).

For pretrained language models, the READ prediction head is the original pretrained LM head inside $f_\theta$. When separate

WRITE/READ heads are used, the WRITE head is initialized from the same pretrained LM head and then fine-tuned for the reconstruction-based WRITE objective. At inference time, both heads and the rest of the model are frozen; only the memory state $\mathcal{M}$ is updated during WRITE. For language modeling experiments, GradMem is trained without a separate WRITE head, as indicated in Table 5.

*Table 5.* Task-specific memory hyperparameters and training initialization. In experiments ARMT uses $d_{\text{armt\_mem}} = 64$ for associative matrix memory. Memory tokens in RMT, ARMT, and GradMem have the same dimension as models input embeddings.

| Task | num_mem_tokens | Training technique |
|---|---|---|
| Associative retrieval (AR) | 8 | Curriculum learning by number of KV-pairs (4, 8, 16, ...); from randomly init model (4 layers, 4 heads, 128 hid) |
| Short SQuAD | 32 | From pretrained model; GradMem trained without learned memory projection |
| bAbI | 8 | From pretrained model |
| Language modeling | 32 | From pretrained model (all models); GradMem trained without separate WRITE head; |

*Table 6.* **Hyperparameters.** "–" indicates the parameter is not applicable. All transformer-based models for associative retrieval (AR) use 4 layers, hidden size 128, 4 attention heads. For NLP tasks, GPT-2 (124M), Pythia (160M), and Mamba (130M) are used as base models. $K$ denotes the number of WRITE gradient steps (GradMem) or forward memory updates (RMT). $\alpha$ is the inner learning rate for GradMem.

| Task | Model | Base LM | Layers | Hidden dim | Heads | # mem tokens ($m$) | $d_{\text{mem}}$ | $K$ | $\alpha$ | Initialization, curriculum | Total memory state size |
|---|---|---|---|---|---|---|---|---|---|---|---|
| Associative retrieval (AR) | Transformer | Llama | 4 | 128 | 4 | – | – | – | – | Random | – |
| | Mamba | Mamba | 4 | 128 | – | – | 16 (state_size) | – | – | Random | 40,960 |
| | ARMT | Llama | 4 | 128 | 4 | 8 | 64 | – | – | Random, 1-2-4-8-16-32-64-96 | 198,144 |
| | RMT | Llama | 4 | 128 | 4 | 8 | – | 1–5 | – | Random, 4-8-16-32-64-96 | 1,024 |
| | TTT-Linear | TTT-Linear | 4 | 128 | 4 | – | – | 1 | 1.0 | Random | 16,896 |
| | LaCT | LaCT | 4 | 128 | 4 | – | – | 1 | 0.001 | Random | 49,152 |
| | GradMem | Llama | 4 | 128 | 4 | 8 | – | 1–5 | [0.01, 10] | Random, 32-64-96 pairs | 1,024 |
| bAbI (QA1–QA5) | GPT-2 | GPT-2 (124M) | 12 | 768 | 12 | – | – | – | – | Pretrained | – |
| | Pythia | Pythia (160M) | 12 | 768 | 12 | – | – | – | – | Pretrained | – |
| | Mamba | Mamba (130M) | 24 | 768 | – | – | 16 (state_size) | – | – | Pretrained | 737,280 |
| | ARMT | GPT-2 (124M) | 12 | 768 | 12 | 8 | 64 | – | – | Pretrained | 3,543,552 |
| | RMT | GPT-2 (124M) | 12 | 768 | 12 | 8 | – | 1 | – | Pretrained | 6,144 |
| | GradMem | GPT-2 (124M) | 12 | 768 | 12 | 8 | – | 1–2 | 0.4 | Pretrained | 6,144 |
| Short SQuAD | GPT-2 | GPT-2 (124M) | 12 | 768 | 12 | – | – | – | – | Pretrained | – |
| | Pythia | Pythia (160M) | 12 | 768 | 12 | – | – | – | – | Pretrained | – |
| | Mamba | Mamba (130M) | 24 | 768 | – | – | 16 (state_size) | – | – | Pretrained | 737,280 |
| | ARMT | GPT-2 (124M) | 12 | 768 | 12 | 32 | 64 | – | – | Pretrained | 3,543,552 |
| | RMT | GPT-2 (124M) | 12 | 768 | 12 | 32 | – | 1 | – | Pretrained | 24,576 |
| | GradMem | GPT-2 (124M) | 12 | 768 | 12 | 32 | – | 1, 5 | 0.4 | Pretrained | 24,576 |
| Language modeling (WikiText-103) | GPT-2 | GPT-2 (124M) | 12 | 768 | 12 | – | – | – | – | Pretrained | – |
| | Pythia | Pythia (160M) | 12 | 768 | 12 | – | – | – | – | Pretrained | – |
| | Mamba | Mamba (130M) | 24 | 768 | – | – | 16 (state_size) | – | – | Pretrained | 737,280 |
| | ARMT | GPT-2 (124M) | 12 | 768 | 12 | 32 | 64 | – | – | Pretrained | 3,543,552 |
| | RMT | GPT-2 (124M) | 12 | 768 | 12 | 32 | – | 1 | – | Pretrained | 24,576 |
| | GradMem | GPT-2 (124M) | 12 | 768 | 12 | 32 | – | 1–2 | 0.4 | Pretrained | 24,576 |

## C. Accelerating Double Backwards Through Attention

In our meta-learning setting, the dominant computational challenge is that the inner-loop optimization requires backwards-over-backwards through attention, which substantially increases both runtime and GPU memory compared to standard training. We implement an efficient double-backward for attention that significantly reduces this overhead on longer sequences: for $L$=1024 tokens, backward time drops from ∼1000 ms to ∼600 ms and peak GPU memory from ∼60 GB to ∼30 GB in our setup. This improvement is critical for scaling GradMem to longer contexts and larger WRITE step counts. In order to accelerate our experiments, we evaluated a few approaches to make the double backward of GradMem more efficient:

- **Eager**: a baseline which fully relies on PyTorch's autograd for first- and second-order differentiation.

- **Fast forward → manual backward**: the forward pass is computed using PyTorch's SDPA kernel. The first-order backward is written analytically, and the second-order derivatives are obtained by differentiating the analytical backward with autograd.

- **Fast forward → autograd**: the forward pass also uses SDPA, but constructs the backward by recomputing the attention forward inside the backward and letting autograd differentiate it, enabling second-order derivatives without storing forward intermediates at the cost of recomputation.

- **Manual HVP**: a fully analytical implementation of forward, backward, and double backward in pure PyTorch.

- **Flash HVP**: fused forward and backward kernels, combined with an analytical double backward.

We compare the speed of backward methods in Figure 5. While on shorter sequences eager attention is the most practical solution, longer contexts benefit from our optimizations, with Fast forward → autograd being the fastest and Manual HVP by far the most memory-efficient. In general, Flash HVP is the most balanced approach, coming in second in both speed and memory requirements.

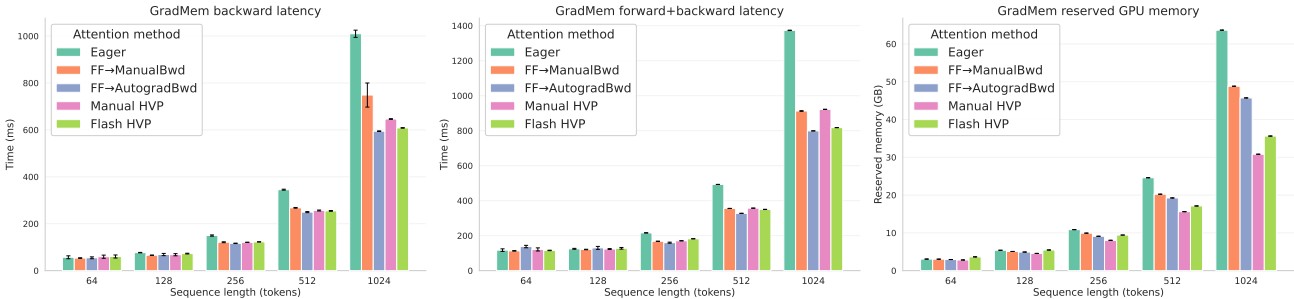

*Figure 5.* **Comparison of speed and memory consumption for attention backwards-over-backwards in GradMem.** These results were obtained using an A100 GPU, with GPT-2 as the base model for GradMem with 8 memory tokens, a query size of 24, 1 inner SGD step and a batch size of 16.

## D. Computational Analysis: When GradMem is Compute-Efficient

GradMem introduces additional WRITE-time compute (test-time optimization) in exchange for cheaper READ-time inference, since subsequent queries attend only over a short memory prefix rather than the full context. Here we characterize when this trade-off is favorable.

Let $c$ be the context length, $q$ the query length with $q \ll c$ is negligible, $m$ the number of memory tokens, $N$ the number of queries asked about the *same* context, and $K$ the number of gradient updates in the WRITE phase. We use $R$ to denote the ratio between the cost of one memory update step and the cost of one forward pass over the context. We compare (i) standard full-context transformer inference that reuses the context for each query, and (ii) GradMem, which pays a WRITE cost once and then answers queries using only memory.

For a transformer-style model, self-attention over a sequence of length $L$ costs $\mathcal{O}(L^2)$, and cross-attention from a query of length $q$ to a context of length $c$ costs $\mathcal{O}(cq)$. If we cache the context representations, standard transformer inference incurs a one-time cost to process $C$, plus a per-query cross-attention cost:

$$T_{\text{full}} \approx c^2 + cqN. \tag{9}$$

For GradMem, the WRITE phase runs $K$ gradient descent steps on the context. Each step costs $R$ times a forward pass over $C$, giving a WRITE cost of $Rc^2K$. At READ time, we process $m$ memory tokens once and each query attends only over the memory tokens and query $q$ tokens. In total giving following cost of both WRITE and multiple reads:

$$T_{\text{GradMem}} \approx R(c+m)^2K + m^2 + mqN. \tag{10}$$

**Break-even condition.** GradMem is compute-efficient when the total cost is lower than full-context inference: $T_{\text{full}}/T_{\text{GradMem}} > 1$. Equivalently,

$$N > \frac{c^2(RK-1) + (1+RK)m^2 + 2cmRK}{q(c-m)}. \tag{11}$$

In the regime where $q$ is treated as a small constant factor, this reduces to the simpler heuristic threshold $N \gtrsim \big(c(RK-1)\big)/q$, which matches the form used in our empirical discussion.

Thus, when the same context is reused for more than the threshold in Equation (11) and $c > m$, GradMem yields lower total compute than repeatedly answering from the full context (Figure 6). In practice, the regime $c \gg m$ and large $N$ (many queries per context) is the most favorable for GradMem, since the amortized savings in READ grow linearly with $N$ while the WRITE cost is paid only once per context.

**Real-use READ/WRITE latency.** On a GPU, models can be bound by factors other than theoretical model complexity, i.e. memory bandwidth or kernel efficiency. In order to control for those factors, we evaluate GradMem against GPT-2 (124M) and Mamba-130m baselines in terms of measured READ and WRITE operation latencies. We treat building the KV-cache in GPT-2 and the recurrent state in Mamba as WRITE operations, while subsequent forward passes using the caches correspond to READ operations.

Figure 7 reports the total latency of a WRITE phase followed by $N$ READ (subsequent queries to the same context) operations from the cached representations for contexts of length 64, 256 and 1024 tokens. GradMem has a large initial cost due to the complexity of gradient-based WRITE, which appears as a higher initial offset. However, the complexity of repeating the READ phase is smaller for GradMem compared to other models, since the underlying transformer attends only to a small set of memory tokens instead of the entire context. In contrast, GPT-2 must repeatedly process its KV-cache, resulting in a higher latency per query.

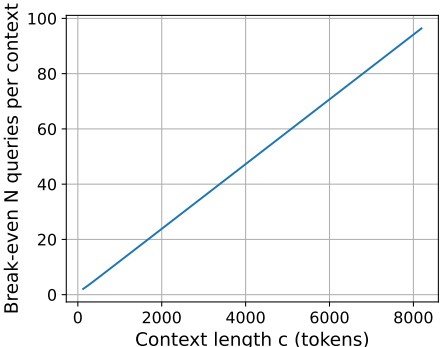

*Figure 6.* **Break-even number of queries for GradMem compute efficiency.** The curve shows the minimum number of queries per context $N$ required for GradMem to use less total compute than full-context inference with cached context representations. Estimates use query length $q$=128 and memory size $m$=32; points above the curve favor GradMem.

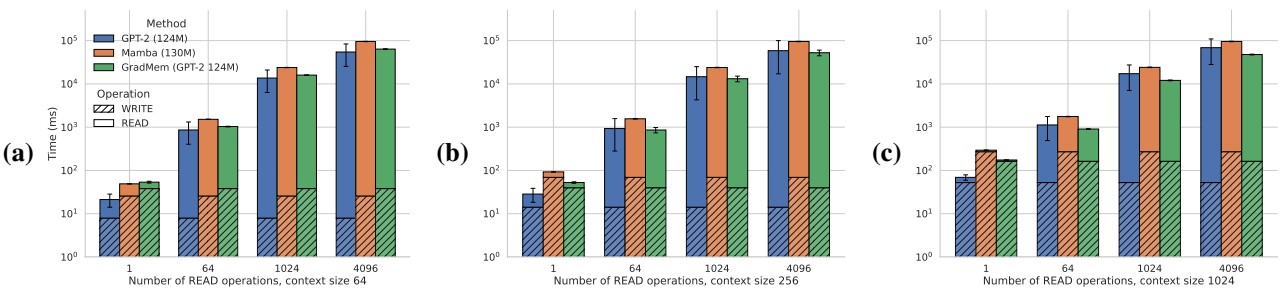

*Figure 7.* **GradMem is competitive in latency when the same context is reused across many READ operations, and becomes about 1.6x faster than Mamba.** Results obtained on an A100 GPU. WRITE+READ latency is shown for contexts of length 64 (a), 256 (b), and 1024 (c) tokens. The query length is 24 tokens and the batch size is 16. GradMem uses GPT-2 as the base model with $K = 1$. For GPT-2 and Mamba-130M, the WRITE operation corresponds to building the KV-cache and the recurrent state, respectively. The y-axis uses a logarithmic scale; the absolute latency difference (ms) increases with the number of queries per context.

As the number of READ operations rises, the initial overhead of GradMem WRITE operation becomes less pronounced. GradMem consistently outperforms Mamba across all evaluated context lengths, and breaks even with GPT-2 after approximately 64 READ phases for the same context (for context size 256 and 1024).

We further evaluate efficiency and break-even points across larger models and longer contexts. The Figure 8 reports total latency for one WRITE followed by multiple READs from the same written context. We use official fast implementations for TTT-Linear, TTT-MLP, and LaCT. For a single READ after WRITE, LaCT / TTT-MLP / TTT-Linear are generally faster than GradMem, since they avoid backpropagation through the full model. However, GradMem amortizes better when the same context is reused across multiple queries: it performs one model-level WRITE per context and then READs from a small fixed-size memory, whereas TTT-style methods perform many token- or chunk-level updates during context processing. At context length 4096, GradMem with Llama-3.2-1B becomes more efficient than LaCT-760M and TTT-Linear-1.3B after 5 and 10 READs, respectively, and has a faster WRITE phase than TTT-MLP-1.3B. For Llama-3-8B, GradMem WRITE+READ takes 1870 / 4860 / 14400 ms at 4k / 8k / 16k tokens, compared to 364 / 747 / 1620 ms for standard Llama-3-8B prefill+decode. In this setup, GradMem becomes more efficient after approximately 27 / 43 / 61 repeated READs from the same context. The reason is that GradMem's READ cost depends on the fixed number of memory tokens, whereas decoding from a Transformer KV-cache grows with context length. For example, Llama-3-8B takes 49.1 ms to decode from an 8k-token cache and 76.6 ms from a 16k-token cache, while GradMem decodes from memory in about 23 ms across these lengths. We use an Nvidia A100 GPU for these experiments.

## E. Relation between Exact Match and Inner Loss

To better understand why extrapolating the number of WRITE iterations improves downstream performance, we analyze the behavior of the inner objective during the WRITE phase. Figure 9 shows that increasing $K$ yields a reduction in the inner loss, indicating that additional WRITE iterations produce a more accurate memory state under the reconstruction objective. Moreover, the reduction in inner loss correlates with improvements in Exact Match when evaluating with larger $K_{eval}$. These results provide evidence that improved context retention—as measured by the inner objective—translates into better task-level accuracy in the context-removal setting.

We further analyze *what* information is being retained by decomposing the reconstruction loss over context tokens into contributions from **key** tokens and **value** tokens in the associative retrieval data. Notably, the loss on key tokens remains comparatively stable across WRITE iterations, while the loss on value tokens decreases as $K_{eval}$ increases. This behavior suggests that the learned memory is *selective*: rather than attempting to store the full context verbatim, the model primarily refines those parts of the representation that are useful for answering queries, i.e., the values that must be produced at READ time. In this regime, keys function mainly as retrieval cues, so improving value reconstruction only is sufficient for increasing Exact Match. This supports two conclusions: (i) better memory fidelity under the inner objective leads to higher downstream accuracy, and (ii) the memory mechanism learns to allocate representational capacity toward answer-relevant content, producing a structured compression in which values are retained more precisely than the keys that index them.

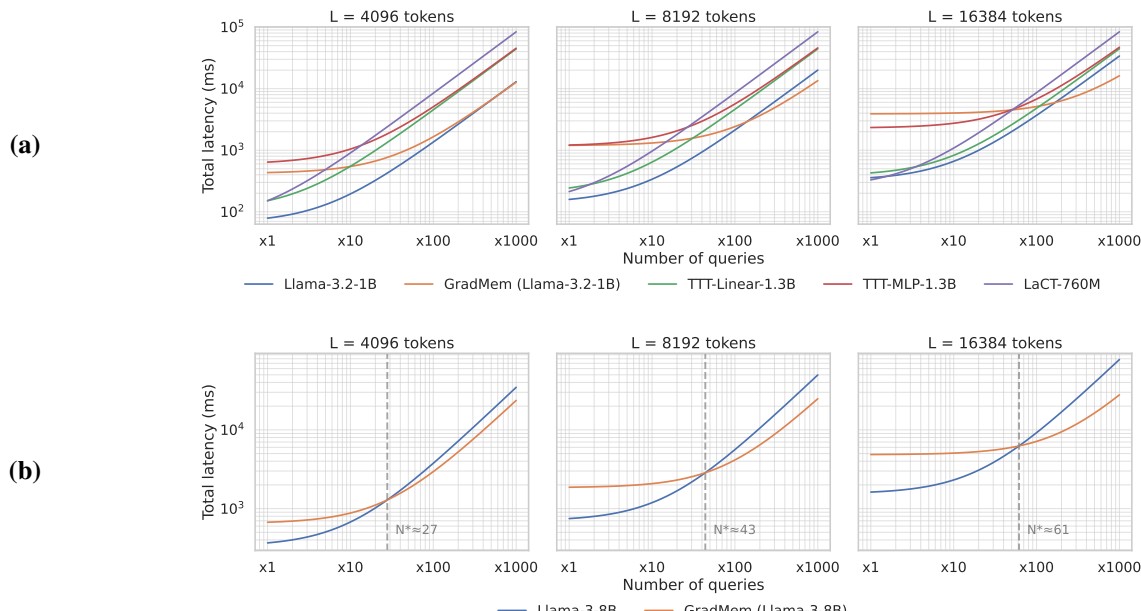

*Figure 8.* **GradMem can be efficiently scaled to longer contexts and larger models.** We benchmark GradMem against **(a)** Llama-3.2-1B and **(b)** Llama-3-8B chosen as its base transformer model, as well as (a) TTT-Linear, TTT-MLP and LaCT. After repeated queries to its memory tokens, GradMem achieves lower overall latency compared to transformers decoding from full KV cache and test-time training methods reusing their state. Query length is 24 tokens, batch size is 1, GradMem has 8 memory tokens and $K = 1$, measurements obtained on A100 GPU.

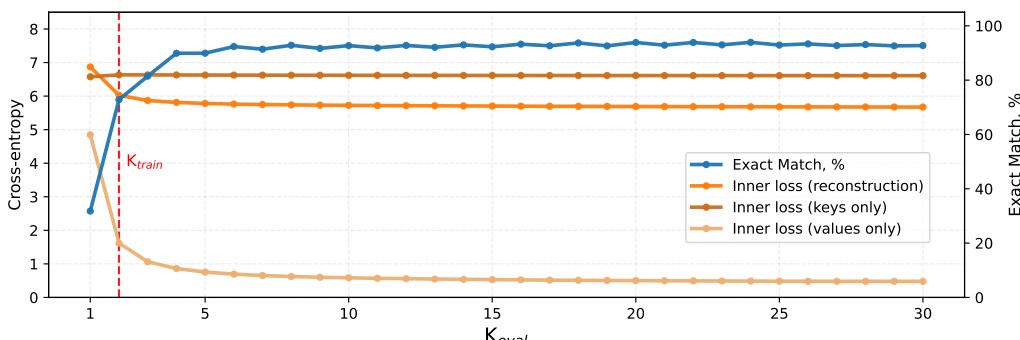

*Figure 9.* **On KV-retrieval, GradMem learns to decrease inner loss on value tokens only.** This figure was obtained for GradMem fine-tuned on 64-pair KV-retrieval with $K_{\text{train}} = 2$.

## F. Memory Size vs. WRITE Steps (K) vs. Number of KV-pairs

We further study the trade-off between memory size, the number of WRITE optimization steps, and the amount of information that must be stored in memory. In particular, we consider associative retrieval with 8 and 16 KV-pairs, sweep the number of memory tokens $m \in \{1, 2, 4, 8\}$, and compare $K = 1$ and $K = 2$ WRITE steps. We start from the checkpoint trained with 8 memory tokens and continue training after reducing the memory to the target size $m$, using the first $m$ vectors from the checkpoint.

Figure 10 shows two clear trends. First, increasing the number of WRITE steps consistently improves Exact Match accuracy for a fixed memory budget. This effect is especially pronounced when memory is small: for example, with 8 KV-pairs, moving from $K = 1$ to $K = 2$ substantially improves performance for $m = 1$ and $m = 2$, and with 16 KV-pairs the gain remains large across the entire low-memory regime. Second, decreasing the number of memory tokens reduces the amount of context that can be reliably stored. As the number of KV-pairs increases from 8 to 16, the same memory budget yields lower Exact Match accuracy, and the performance curves shift to the right.

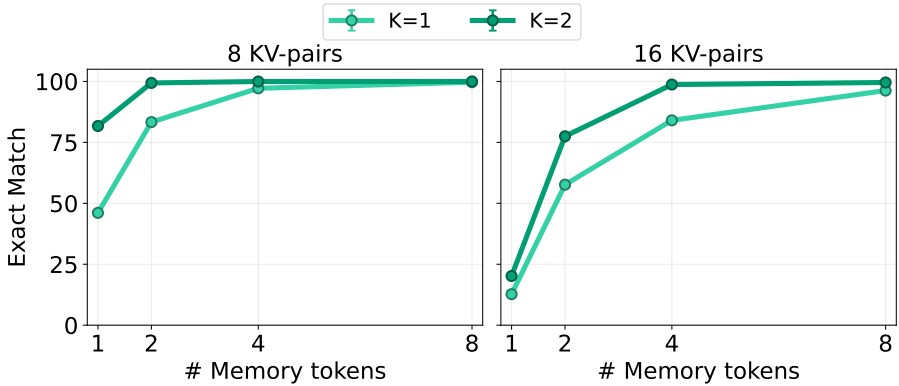

*Figure 10.* **More WRITE steps improve memory efficiency, but larger contexts still require larger memory.** Exact Match accuracy on associative retrieval as a function of the number of memory tokens $m \in \{1, 2, 4, 8\}$ for two numbers of WRITE steps, $K \in \{1, 2\}$, and two context sizes: 8 and 16 KV-pairs. All runs are initialized from the checkpoint trained with 8 memory tokens and then further trained after reducing the memory size. Increasing $K$ consistently shifts the curve upward, especially with smaller memory size, but the memory size requirement still grows with the number of KV-pairs: for 8 KV-pairs, $K = 2$ reaches near-perfect performance already with $m = 2$, whereas for 16 KV-pairs near-perfect performance requires at least $m = 4$.

These results suggest that $K$ can partially compensate for a smaller memory by making WRITE more effective, but only up to a point: when memory is too small, additional optimization steps do not fully remove the bottleneck. In our setup, 8 KV-pairs can already be stored almost perfectly with only $m = 2$ memory tokens when using $K = 2$, whereas 16 KV-pairs require a larger memory budget, with near perfect performance reached only at $m \geq 4$. Overall, more WRITE steps improve memory utilization, but larger contexts still require larger memory states.

## G. Memory State Size Utilization: GradMem vs. Mamba

Compared to Mamba, GradMem performs compression much better, when the state size is limited. In our main experiments we reuse most Mamba parameters from mamba-130m checkpoint, which results in much larger memory state size (around 41k floats across all layers) compared to GradMem (1024 floats across 8 mem tokens). Here we compare multiple ways to match Mamba memory state with GradMem by fixing the hidden state size ($d\_model = 128$) and experiment with $conv\_kernel$ (conv) and $state\_size$ (state). As shown in Table 7, with these limited constraints, GradMem significantly outperforms Mamba even when the latter has $16\times$ larger total state.

*Table 7.* Associative retrieval performance, exact match (%) (mean $\pm$ std) across task lengths. All models have the same hidden state size of 128, the results are averaged across 3 runs.

| Model | Config | Total State Size | N8 | N16 | N32 |
|---|---|---|---|---|---|
| Mamba | 1L state 4 conv 4 | 4096 | $69.2 \pm 6.4$ | $56.8 \pm 6.9$ | $12.4 \pm 15.5$ |
| Mamba | 2L state 2 conv 2 | 4096 | $60.6 \pm 41.2$ | $15.9 \pm 9.1$ | $4.9 \pm 6.2$ |
| Mamba | 4L state 4 conv 4 | 16384 | $96.6 \pm 0.5$ | $82.9 \pm 1.6$ | $19.8 \pm 19.3$ |
| Mamba | 4L state 16 conv 4 | 40960 | $98.8 \pm 1.5$ | $98.7 \pm 0.8$ | $90.4 \pm 4.7$ |
| GradMem | 4L k=1 | 1024 | $99.7 \pm 0.0$ | $96.3 \pm 0.9$ | $86.9 \pm 0.5$ |
| GradMem | 4L k=2 | 1024 | $100.0 \pm 0.0$ | $99.6 \pm 0.1$ | $98.3 \pm 0.1$ |
| GradMem | 4L k=5 | 1024 | $100.0 \pm 0.0$ | $100.0 \pm 0.0$ | $99.9 \pm 0.1$ |

## H. KV Retrieval with LaCT and TTT-Linear

To compare GradMem with prior test-time-training methods that adapt layer-local state during context processing, we evaluate TTT-Linear (Sun et al., 2025) and LaCT (Zhang et al., 2026) on the same associative KV-retrieval data and model scale as in Table 1. Unlike GradMem, which optimizes a single input-level memory state using a model-level context-reconstruction objective in a dedicated WRITE phase, TTT-Linear and LaCT maintain layer-local fast-weight/state variables and update them with local self-supervised objectives while processing the context (token/mini-batch updates for TTT-Linear and large-chunk updates for LaCT). Because input-level memory in GradMem is supplied as prefix tokens, it

preserves the base architecture and requires no internal layer changes, per-layer writable states, or layer-local objectives. This comparison is not matched by memory state size: in this setup LaCT uses about 49k memory-state floats and TTT-Linear about 17k, whereas GradMem uses 1024 floats across 8 memory tokens for 4-layer 128-hidden models (see Table 6 for hyperparameter details).

*Table 8.* **KV-retrieval comparison with TTT-Linear and LaCT.** Exact match retrieval accuracy (mean±std over 3 runs unless otherwise noted) on the same associative KV-retrieval setup as Table 1. LaCT has high variance, so we also report the best of 3 runs.

| Model | \multicolumn{5}{c}{Number of KV-pairs} |
|---|---|---|---|---|---|
| | 8 | 16 | 32 | 64 | 96 |
| TTT-Linear | $97.2_{\pm0.9}$ | $96.9_{\pm0.7}$ | $82.1_{\pm3.4}$ | $38.6_{\pm14.4}$ | $11.9_{\pm10.8}$ |
| LaCT | $95.7_{\pm6.8}$ | $99.9_{\pm0.1}$ | $99.7_{\pm0.4}$ | $62.5_{\pm25.2}$ | $46.9_{\pm24.3}$ |
| LaCT (best of 3) | 99.9 | 99.9 | 99.9 | 91.6 | 74.9 |
| GradMem ($K=1$) | $99.7_{\pm0.0}$ | $96.3_{\pm0.9}$ | $86.9_{\pm0.5}$ | $58.6_{\pm0.7}$ | $32.6_{\pm0.1}$ |
| GradMem ($K=5$) | $\mathbf{100.0}_{\pm0.0}$ | $\mathbf{100.0}_{\pm0.0}$ | $\mathbf{99.9}_{\pm0.1}$ | $\mathbf{99.1}_{\pm0.3}$ | $\mathbf{88.4}_{\pm2.3}$ |

LaCT is a strong baseline but has high variance, especially at larger numbers of key–value pairs. GradMem with one WRITE step is comparable to TTT-Linear and below the best LaCT runs at larger context sizes, while GradMem with additional WRITE steps remains strongest overall despite using a much smaller model-level memory state.

## I. Text Compression with GradMem on Larger Pretrained Models

To test whether GradMem is applicable beyond GPT-2-scale models, we study a text reconstruction task with pretrained Llama-3.2-1B and Llama-3.2-3B models. In terms of our WRITE/READ formulation, a text segment is provided as context $C$, compressed into memory $M$ during WRITE, and then reconstructed from $M$ during READ using the query `text:`. We measure token-level reconstruction accuracy.

During WRITE, we apply learned LoRA adapters to the model to optimize only the memory vectors at test time. During READ, the pretrained base model is frozen and used without LoRA, i.e., we do not modify the model at READ itself. This shows that GradMem can be trained so that a pretrained large model can directly consume the learned memory state at inference time. We train models for each configuration of sequence length $N$, number of WRITE steps $K$, and number of memory vectors $n_{\mathrm{mem}}$. The data comes from PG19 (Rae et al., 2020), and the token-level accuracy of base models on these texts is about 10–15%, making the reconstruction task non-trivial.

Table 9 reports the maximum sequence length $N$ that reaches 95% and 99% token accuracy on a coarse grid $N \in \{8, 16, 32, 64\}$. The main conclusion is that Grad-Mem remains effective on 1B- and 3B-parameter pretrained models even in a highly compressed regime with only 1–2 memory vectors. Increasing the number of WRITE steps from $K = 1$ to $K = 2$ consistently improves the amount of text that can be reliably stored, mirroring the trend we observe in KV retrieval when varying memory size and WRITE compute. For example, on Llama-3.2-1B, moving from $K = 1$ to $K = 2$ increases $N@95$ from 8 to 32 with $n_{\mathrm{mem}} = 1$, and from 16 to 32 with $n_{\mathrm{mem}} = 2$, while also improving $N@99$ up to 32 for $n_{\mathrm{mem}} = 2$. On Llama-3.2-3B, $K = 2$ reaches $N@99 = 32$ already with a single memory vector, and reaches $N@95 = 64$ with two memory vectors.

Overall, these results show that GradMem is not restricted to 100M-scale models. It can be trained and applied on 1–3B pretrained LLMs, can store text segments in as little as 1–2 memory vectors with 1–2 WRITE steps, and does so while leaving the READ model unchanged. This

*Table 9.* **GradMem is applicable to larger pretrained LLMs: with only 1–2 memory vectors and 1–2 WRITE steps, it can compress and reconstruct short text spans while keeping the READ model frozen.** Maximum sequence length $N$ that reaches target token-accuracy thresholds on text reconstruction with GradMem. Higher is better. We evaluate Llama-3.2-1B and Llama-3.2-3B with $K \in \{1, 2\}$ WRITE steps and $n_{\mathrm{mem}} \in \{1, 2\}$ memory vectors. During WRITE, the model uses learned LoRA adapters, while during READ the pretrained base model is frozen and used without LoRA. Sequence lengths are evaluated on a coarse grid $N \in \{8, 16, 32, 64\}$, so the reported thresholds should be interpreted as approximate capacity estimates.

| Model | $K$ | $n_{\mathrm{mem}}$ | $N@95$ | $N@99$ |
|---|---|---|---|---|
| Llama-3.2-1B | 1 | 1 | 8 | 8 |
| | 1 | 2 | 16 | 8 |
| | 2 | 1 | **32** | 16 |
| | 2 | 2 | **32** | **32** |
| Llama-3.2-3B | 1 | 1 | 16 | 8 |
| | 1 | 2 | 16 | 16 |
| | 2 | 1 | 32 | **32** |
| | 2 | 2 | **64** | **32** |

corresponds to roughly 16x–32x compression, since up to 32–64 input tokens are stored in only 1–2 memory vectors. Since the sequence-length sweep is coarse, the values in Table 9 should be interpreted as approximate capacity estimates.

## J. Input-level vs. Per-layer Memory

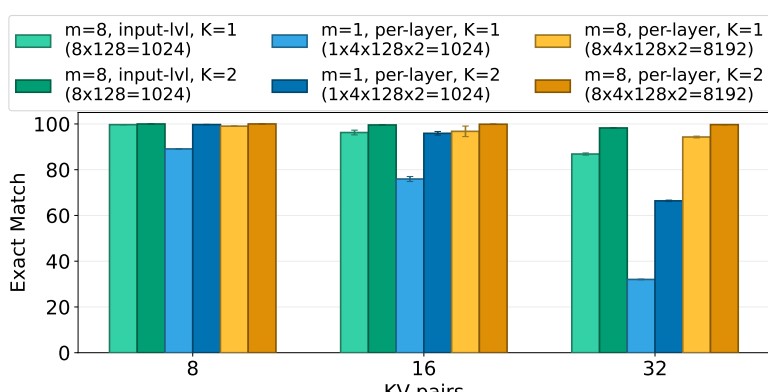

*Figure 11.* **Input-level memory is more parameter-efficient than per-layer memory, while a much larger per-layer state can outperform it.** Exact Match on associative KV retrieval for 8/16/32 KV-pairs using 4-layer models with hidden size 128. We compare GradMem's input-level memory with 8 memory tokens ($8 \times 128 = 1024$ parameters) against per-layer memory implemented as test-time trainable KV-cache in two regimes: (i) matched total memory size, corresponding to KV-cache for one token across all 4 layers ($1 \times 4 \times 128 \times 2 = 1024$), and (ii) matched KV-cache length for 8 tokens ($8 \times 4 \times 128 \times 2 = 8192$), which is $8\times$ larger. All variants use the same model-level reconstruction objective; only the parameterization of test-time trainable memory changes. With matched memory size, input-level memory is consistently stronger, especially at longer contexts; with the much larger per-layer state, per-layer memory becomes strongest.

We compare the *input-level* memory used by GradMem to a *per-layer* memory parameterization in which each layer has its own test-time trainable KV-cache state. The goal of this ablation is to isolate the effect of *where* memory is stored. Importantly, the training objective remains the same in all cases: a model-level reconstruction loss on associative KV retrieval. We only change the set of test-time trainable memory parameters.

All experiments use 4-layer models with hidden size 128 and are evaluated on associative retrieval with 8, 16, and 32 KV-pairs. GradMem uses 8 input memory tokens, for a total memory size of $8 \times 128 = 1024$ parameters. We compare this to two per-layer memory setups: (1) a *size-matched* configuration with the same total number of memory parameters, corresponding to a KV-cache for one token across all layers, $1 \times 4 \times 128 \times 2 = 1024$; and (2) a larger per-layer memory with the same KV-cache length as 8 memory tokens, $8 \times 4 \times 128 \times 2 = 8192$, i.e. $8\times$ more parameters. We report results for $K = 1$ and $K = 2$ WRITE steps.

Figure 11 shows that, when memory size is matched, input-level memory is consistently better than per-layer memory, and the gap grows as the task becomes harder. For example, at 32 KV-pairs, the size-matched per-layer memory is far below input-level memory for both $K = 1$ and $K = 2$. At the same time, if the per-layer KV-cache is allowed to use a much larger state ($8\times$ more parameters), it becomes the strongest variant. This suggests that input-level memory is more parameter-efficient: a compact set of input memory tokens can still be transformed by the model into richer internal KV-cache representations at higher layers, whereas directly optimizing per-layer KV states only becomes advantageous when given substantially larger capacity.

## K. GradMem Algorithm and Minimal Code

Algorithm 1 gives the mathematical WRITE/READ procedure. Below it, we include a minimal PyTorch-like version that mirrors our implementation while omitting padding, separate WRITE heads, and other engineering details. The key points are that WRITE takes gradients only with respect to the per-example memory, READ removes the original context, and training backpropagates the outer loss through the WRITE updates. Full implementation is available at https://github.com/yurakuratov/gradmem.

1: **WRITE: encode context into memory**
**Require:** Context $C = (t_1, \ldots, t_N)$, meta-learned initialization $\mathcal{M}_0$, model $f_\theta$ not updated during WRITE, WRITE steps $K$, learning rate $\alpha$
2: $\mathcal{M} \leftarrow \mathcal{M}_0$         $\triangleright$ Initialize from meta-learned state
3: **for** $k = 1, \ldots, K$ **do**
4:     $\mathcal{L}_{\text{write}}(\mathcal{M}; C) \leftarrow -\sum_{i=1}^{N} \log f_\theta(t_i \mid [\mathcal{M}; t_{<i}])$         $\triangleright$ Reconstruction loss
5:     $\mathcal{M} \leftarrow \mathcal{M} - \alpha \nabla_{\mathcal{M}} \mathcal{L}_{\text{write}}(\mathcal{M}; C)$         $\triangleright$ Gradient descent on memory only
6: **end for**
7: **return** $\hat{\mathcal{M}} \leftarrow \mathcal{M}$
8:
9: **READ: predict from memory and query**
**Require:** Optimized memory $\hat{\mathcal{M}}$, query $Q$, model $f_\theta$
10: $\hat{Y} \leftarrow f_\theta(\cdot \mid [\hat{\mathcal{M}}; Q])$         $\triangleright$ Decode without access to $C$
11:
12: **Meta-training: outer loop, not run at test time**
13: $\mathcal{L}_{\text{task}} \leftarrow -\log f_\theta(Y \mid \hat{\mathcal{M}}, Q)$
14: Update $\theta, \mathcal{M}_0$ by $\nabla_{\theta, \mathcal{M}_0} \mathcal{L}_{\text{task}}$         $\triangleright$ Backprop through WRITE steps

*Algorithm 1.* GradMem WRITE/READ procedure and outer-loop meta-training.

```python
import torch

def write(model, M0, context_ids, K, alpha, train=False):
    # One memory copy per example. Only M is updated in WRITE.
    B = context_ids.size(0)
    M = M0[None].expand(B, -1, -1).clone().requires_grad_(True)

    for _ in range(K):
        x = prepend_memory(model, M, context_ids)
        loss = context_reconstruction_loss(model, x, context_ids)

        # train=True keeps the graph for the second-order meta-gradient.
        (gM,) = torch.autograd.grad(loss, M, create_graph=train)
        M = M - alpha * gM

        # At test time, keep only the new memory value, not the graph.
        if not train:
            M = M.detach().requires_grad_(True)
    return M if train else M.detach()

def read(model, M, query_ids):
    x = prepend_memory(model, M, query_ids)
    return model(inputs_embeds=x).logits  # no original context

def train_step(model, M0, batch, optimizer, K, alpha):
    M = write(model, M0, batch.context_ids, K, alpha, train=True)
    logits = read(model, M, batch.query_ids)
    loss = target_loss(logits, batch.labels)  # outer READ loss

    optimizer.zero_grad()
    loss.backward()     # updates model/meta-params through WRITE
    optimizer.step()
    return loss

def predict(model, M0, context_ids, query_ids, K, alpha):
    # Test-time WRITE still uses gradients, but only to update memory.
    M = write(model, M0, context_ids, K, alpha, train=False)
    with torch.no_grad():
        return read(model, M, query_ids)
```

*Figure 12.* Minimal PyTorch-like GradMem code. Helper functions hide tokenization, padding, label shifting, and memory-prefix construction.

