# OpenReview forum: "GradMem: Learning to Write Context into Memory with Test-Time Gradient Descent"
_ICML.cc/2026/Conference — ICML 2026 regular_

### Official Review · Reviewer_wzEW · 2026-03-08

**Soundness:** 4
**Presentation:** 4
**Significance:** 2
**Originality:** 3
**Overall Recommendation:** 5
**Confidence:** 4

**Summary:**

This paper studies memory compression for Transformer language models in a memory removal setting, where the model must generate answers without access to the full context. The authors propose GradMem, a MAML-based method that learns a set of memory embedding tokens representing compressed context.
Starting from a meta-learned initialization, the memory tokens are updated via a small number of gradient steps using a reconstruction loss in an inner optimization loop. After K gradient steps, the updated memory tokens are prepended to the input prompt and used to generate answers without the original context. The outer-loop training objective meta-learns the memory initialization by backpropagating through the inner-loop updates using next-token cross-entropy on the reference answer.
Experiments include pretraining small 4-layer transformers on synthetic key-value retrieval tasks and finetuning ~100M parameter GPT-2 models on natural language tasks such as bAbI and SQuAD variants. GradMem consistently improves with additional inner-loop updates and outperforms baselines that update memory only through forward passes.

**Compliance With Llm Reviewing Policy:**

Affirmed.

**Final Justification:**

The rebuttal addressed all my questions. I maintain my acceptance score.

**Key Questions For Authors:**

1) Table 1 and Table 2 (GradMem rows with increased K):
 For some tasks, the entries are marked as “–”, whereas GradMem with K=1 can solve the task to some degree. Could the authors clarify why results are missing in these cases? Did you observe divergence or instability in the inner-loop updates?

2) Inner-loop stability:
 Did the authors observe any instability when increasing the number of inner-loop gradient steps? If so, what mechanisms (e.g., gradient clipping, step size tuning) were required to maintain stable updates?

3) RMT repeated forward passes:
 When applying repeated forward passes with RMT, are the memory tokens accumulated across passes (e.g., summed or otherwise aggregated), or replaced at each pass?

4) Scalability:
 Do the authors anticipate practical challenges when applying GradMem to very large language models, given that inference requires gradient computation through the entire model?

5) Context Size vs. Memory Size
The authors use different numbers of memory tokens for the different tasks (Table 4). Did the authors further investigate the Memory Size vs. Context Size (vs. number of gradient steps) tradeoff?


Minor Comments:
L.99: “memoriztion” → “memorization”
L.352: repeated sentence.

**Limitations:**

yes

**Strengths And Weaknesses:**

**Strengths**
- Clear problem formulation and well-defined memory removal setup.
- Excellent Figure 1, which clearly illustrates the method.
- Carefully designed experiments with well-justified baselines.
- Strong experimental structure, making results easy to follow.
- The analysis in Sec. 3.4 shows that increasing the number of inner-loop updates at inference time (K_{eval}​) leads to consistent improvements and extrapolates beyond the number used during training.
- The paper includes speed and memory benchmarks for efficient GradMem implementations.
- Code is provided, which improves reproducibility (although the Flash HVP Triton kernel file appears empty).


**Weaknesses**
- Experiments are conducted at a very small scale (4-layer models and ~100M parameter GPT-2), which limits conclusions about applicability to modern LLMs.
- Additional experiments on scaling memory size (e.g. number of memory tokens) and context size would further deepen the understanding of GradMem.
- Because GradMem updates memory tokens via gradient descent, a backward pass through the entire model is required at inference time, which may be challenging for very large models.
- The limitations and scalability considerations could be discussed more explicitly, particularly regarding whether and how the approach could scale to multi-billion parameter LLMs.
- The paper provides only high-level implementation details in the main text and appendix; additional detail would help reproducibility, though code is included in the supplementary material.


Despite the limited scale, the experiments are well-motivated and carefully executed, and they demonstrate the effectiveness of GradMem in the studied setting.
GradMem is well motivated with observations from recent literature [1]. It elegantly applies a previous meta-learning algorithm (MAML) to the task of context compression into fixed size memory tokens in the domain of language models.
The authors differentiate GradMem in detail from the related test-time training (TTT) layers. To the best of my knowledge GradMem seems novel.

[1] Kuratov, Y., Arkhipov, M., Bulatov, A., and Burtsev, M. Cramming 1568 tokens into a single vector and back again: Exploring the limits of embedding space capacity. 2025

---

> ### Author Rebuttal · Authors · 2026-03-31
>
> Thank you for the careful and supportive review. We especially appreciate that you highlighted the clear problem formulation, the strength of Figure 1, and the well-justified experimental design.
>
> ### On scalability to larger models
>
> We focused on ~100M models because that scale allowed controlled experiments under limited compute. We now add new evidence for scaling: latency measurements up to 8B parameters and new memory-writing experiments up to 3B parameters.
>
> Specifically, we train GradMem on a text compression task with Llama-3.2-1B and Llama-3.2-3B, where a text segment is written into memory and reconstructed from the query “text:”. During WRITE, we use learned LoRA adapters, and during READ, the pretrained base model is used without LoRA. So, the base model is unchanged at READ time. Even with only 1–2 memory vectors and 1–2 WRITE steps, GradMem stores text reliably. On Llama-3.2-1B, K=2 reaches 95% token accuracy on 32-token texts (N@95=32) and N@99=32, while on Llama-3.2-3B, K=2 reaches N@99=32 with one memory vector and N@95=64 with m=2. This corresponds to roughly 16-32x compression and shows applicability to 1–3B pretrained LLMs. Table: (https://postimg.cc/G8TphrbG).
>
> Scaling WRITE to very large models remains challenging because it requires a backward pass through the model. The optimizer state is small, since only memory vectors are optimized at inference, and activation memory can be reduced with gradient/activation checkpointing, but the backward pass remains the main bottleneck. We therefore view GradMem’s current best use case in settings where a context is written once and reused many times. Appendix D analyzed this amortized regime. We now complement it with measured long-context latency for Llama-3-8B at 4k / 8k / 16k tokens. Combined WRITE+READ takes about 1870 / 4860 / 14400 ms, while standard Llama-3-8B prefill+decode takes about 364 / 747 / 1620 ms. Because GradMem’s READ cost is nearly constant while KV-cache decoding grows with context length, GradMem becomes favorable after about 27 / 43 / 61 repeated READs for 4k / 8k / 16k contexts, respectively. For full latency details, please see our response to reviewer oBek (“On latency”).
>
> We will make these scalability limitations more explicit in the revision.
>
> Full measurements for GradMem with Llama-3-8B and Llama-3.2-1B: (https://postimg.cc/3d2QrNmj).
>
> ### On context size, memory size, and gradient steps
>
> Thank you for raising this point. We further investigated this tradeoff and found that increasing K makes a fixed memory budget more effective, but larger contexts still require larger memory. For example, with 8 KV-pairs, K=2 reaches near-perfect performance already at m=2, whereas with 16 KV-pairs even K=2 still requires at least m=4. For the NLP experiments in Table 4, memory sizes were chosen manually rather than through an exhaustive tuning study. The goal there was to test transfer to NLP and compare GradMem to RMT under the same memory size. Full scaling results: (https://postimg.cc/zLBB6K6X).
>
> ### On reproducibility
>
> Thank you, we agree that the implementation details in the paper can be more explicit. We will open-source the code and add both a formal pseudocode description of GradMem and a table of task-specific hyperparameters to the revision. They are provided here: [algorithm](https://postimg.cc/2Vv3pnwT), [hyperparameters](https://postimg.cc/K4C4s0n7).
>
> ### On missing table entries and the effect of larger K
>
> Thank you for pointing this out. We have now completed several of the previously missing runs. Updated [Table 1](https://postimg.cc/d7x1YDcG) and  [Table 2](https://postimg.cc/DJ4zyF7Q). As discussed in reviewer y4zh (“On missing Table 1 entries…”), the effect of larger K is task-dependent, while the LM run with K=2 gives 2.91±0.01, whereas the clearest gain is on SQuAD.
>
> ### On inner-loop stability
>
> Yes, increasing the number of inner-loop WRITE steps can make training less stable, since each additional step deepens the meta-gradient computation. In practice, the most effective stabilization was curriculum learning -- start with shorter sequences and smaller K, then gradually increase the difficulty. We will clarify this in the revision.
>
> ### On repeated forward passes in RMT
>
> In repeated forward passes with RMT, memory is replaced rather than accumulated. As a result, the same fixed WRITE rule is reapplied without an explicit per-example error signal, so extra passes can overwrite or interfere with useful memory content instead of steadily correcting it. This is why extra forward passes give limited or inconsistent gains, whereas additional gradient-based WRITE steps in GradMem are much more reliable.

---

> > ### Author Rebuttal · Reviewer_wzEW · 2026-04-02
> >
> > I maintain my acceptance score.

---

### Official Review · Reviewer_oBek · 2026-03-09

**Soundness:** 2
**Presentation:** 3
**Significance:** 2
**Originality:** 2
**Overall Recommendation:** 3
**Confidence:** 4

**Summary:**

This paper proposes GradMEM, an approach to compress the context of the KV cache into a few memory tokens. The main goal is to avoid full attention computations, as followed comprehensively in existing TTT work [1,2,3,4]. Specifically, a gradient descent step is used to write context into memory conditioned on reconstruction loss of the subsequent token(s). Then, the context can only condition on these tokens and avoid using the full context. A good initialization for the memory $M$ is learnt via meta learning.

**Compliance With Llm Reviewing Policy:**

Affirmed.

**Final Justification:**

I have increased my score to a 3 based on the author's response.

There are certain aspects that I still am not fully convinced by.
For example, LaCT outperforms GradMem (K=1). Given that LaCT does not perform multiple gradient steps, it's unclear how it would behave with multiple steps compared to GradMem.
Furthermore, it seems that the speed gains of GradMem diminishes compared to LaCT with increasing performance as it requires back propagation through the whole model.

**Key Questions For Authors:**

N/A, see main review.

**Limitations:**

Yes

**Strengths And Weaknesses:**

Soundness:
1. The paper is methodologically sound: It is intuitive and easy to understand why training to learn a compressed set of latent vectors of context can help.
2. Missing comparison to methods like Titans/ATLAS, TTT-KVB, TTT-MLP/Linear. The paper does not provide enough experimental results to confirm why their proposed approach advances existing line of literature solving similar goals. It only discusses TTT-MLP/Linear (Sun et al 2025) in terms of methodological difference.
3. It is important to understand and benchmark the latency of this approach against baseline (and existing gradient-based work outlined above) more rigorously. Many long-context line of works in the TTT space mainly care about latency because exact attention computation can be traded off for latency via tiling, like how Flash Attention does. While Appendix C and D discusses how the gradient descent used in this work can be made faster, and when GradMem "breaks-even", it is still not clear how it compares wrt other methods **end-to-end** at inference time, and where it lies on the performance-latency frontier.

Presentation:
1. The overall presentation of the method and additional experiments is good.
2. The tables 1 and 2 are hard to read. I don’t fully understand why some numbers are missing, or how GradMem is considered competitive when it severely underperforms baselines in Table 2.

Significance:
1. In its current form, I believe that this work is an interesting proof of concept, but requires a major round of revision, comparisons, performance improvement over the recurrent baselines, and more experimental results validating the latency utility of this approach.
2. As such, it does not seem evident to me that this work can be immediately be useful and built upon by the community without some revisions.

Originality:
1. Titans (NeurIPS 2025, Dec 2024 preprint)/ATLAS maintain persistent memory + neural memory modules aimed to explicitly compress context - the same goal as what is studied in this paper. It follows the same stage of write into neural memory via gradient descent. The neural memory $M$ here is also a meta-model. Therefore I don’t think the novelty is well justified - or at least clear to me in this form of the paper.

[1] Titans: https://arxiv.org/abs/2501.00663

[2] TTT-KVB: https://arxiv.org/abs/2505.23884

[3] TTT-MLP/Linear: https://arxiv.org/abs/2407.04620

[4] ATLAS: https://arxiv.org/abs/2505.23735

---

> ### Author Rebuttal · Authors · 2026-03-31
>
> Thank you for the detailed review. We appreciate that you found the paper methodologically sound and the overall presentation good, and we take seriously your concerns about empirical positioning relative to recent TTT / neural-memory work and latency benchmark.
> ### On positioning relative to prior TTT and neural-memory work
> We agree that the distinction from prior work [1-4] should be clearer. We provide a direct comparison in the [Table](https://postimg.cc/FYkyBTCs).
>
> The main difference from [1-4] is that GradMem uses a single segment-level memory state, so WRITE and READ are performed once per context at the model level, rather than maintaining separate fast states at each layer. Correspondingly, prior TTT methods mainly write intermediate activations, whereas GradMem aims to write the input context itself into memory. Some prior methods such as [1] and [3] also use token-wise/online updates, while GradMem writes the entire context at once, with memory capacity controlled by the number of memory tokens.
>
> Another key difference is that GradMem allows multiple gradient-descent WRITE steps, rather than a single update. This matters because it lets WRITE benefit from iterative optimization, which is one of the main reasons we observe stronger performance as the number of WRITE steps increases.
>
> To compare with prior methods, we added TTT-linear results on KV retrieval. On 8/16/32/64/96 pairs it achieves 97.2/96.9/82.1/38.6/11.9 EM (3 runs avg), which is still behind GradMem with K=1. Titans and ATLAS do not provide code or checkpoints, making experimental comparison difficult.
>
> We will revise the paper to make this originality claim explicit: GradMem is not just memory written by gradient descent but (1) model-level, (2) context-level, (3) multi-step gradient-based WRITE mechanism with (4) explicitly controllable memory size, (5) trained with second-order meta-learning through the whole model and inner optimization loop.
>
> ### On latency
> Appendix D gave analytical break-even estimates. Additionally, we evaluated GradMem end-to-end latency at much longer contexts with Llama-3-8B at 4k/8k/16k tokens. Combined WRITE+READ latency is 1870/4860/14400ms, compared to 364/747/1620ms for standard Llama-3-8B prefill+decode. In this setup, GradMem becomes more efficient after 27/43/61 repeated READs from the same context on an A100 with bs=1.
>
> The reason is that GradMem READ cost depends on a fixed number of memory tokens, whereas Transformer KV-cache decoding grows with context length. Thus, Llama-3-8B takes 49.1ms to decode from an 8k-token cache and 76.6ms from a 16k-token cache, while GradMem decodes from memory in 23ms. We observe the same trend for smaller models used in the paper, thus GPT-2-124M, GradMem combined WRITE+READ cost becomes smaller than Transformer prefill+decode after 20/50 repeated READs at 256/1024 tokens contexts. At GPT-2 scale, we also report latency against Mamba.
>
> Full results for GradMem: [Llama-3-8B, Llama-3.2-1B](https://postimg.cc/3d2QrNmj), [GPT-2](https://postimg.cc/FdkKsmhL) will be incorporated into revision.
>
> To keep the claim precise, this still does not provide a direct comparison to [1-4], but it does address the main gap by replacing analytical break-even estimates with measured long context latency.
>
> ### On table clarity and competitive results
> We agree that Tables 1&2 in submission were harder to read and interpret. We have now added several of the missing runs, which makes both tables substantially clearer.
>
> In Table 1, we added the 4- and 8-pair GradMem results, as well as RMT results up to 96 KV pairs. GradMem reaches 100% at 4 pairs and 99.7% at 8 pairs, and additional test-time gradient steps recover 100% on both. We also added longer-context RMT results, which show sharp degradation beyond 32 KV pairs: 19.3% at 64 and 12.9% at 96. In Table 2, we filled the previously missing larger K runs. For example, the missing LM run with K=2 gives validation cross-entropy 2.91, only a small improvement relative to K=1 2.92, while larger K has a clearer effect on the SQuAD variant. Updated [Table 1](https://postimg.cc/d7x1YDcG), [Table 2](https://postimg.cc/DJ4zyF7Q).
>
> We will also tighten the wording around "competitive". We do not claim that GradMem outperforms all baselines in Table 2. Our claim is narrower -- the same task-agnostic gradient-based WRITE rule that works strongly on the controlled KV benchmark also transfers beyond that synthetic task, as GradMem achieves >99% on 4/5 bAbI tasks, and on SQuAD it outperforms the matched-memory baseline RMT.
>
> Transformer with full attention should be viewed only as an upper bound. Likewise, Mamba and ARMT are strong baselines, but are not directly matched to GradMem, as they use much larger memory/state, and Mamba's state is updated at every token, whereas GradMem performs only K (1–2) WRITE updates per context. We already revised tables and will revise wording to make these comparison axes explicit and the results easier to interpret.

---

> > ### Author Rebuttal · Reviewer_oBek · 2026-04-01
> >
> > Thank you for the additional experiments which have clarified some of my concerns.
> >
> > Based on the authors' responses, I have the following questions:
> > 1. Given that ARMT and Mamba are strong baselines, I would then like to understand the contextualization of GradMem. Are there settings where GradMem would be preferred over those approaches or vice versa?
> >
> > 2. > TTT-linear results on KV retrieval
> >
> > Was this model trained on the same data?
> >
> > 3. > GradMem allows multiple gradient-descent WRITE steps
> >
> > I consider this a hyperparameter rather than a novelty as in principle the other memory based approaches could do multiple gradient descent steps too. If the authors think otherwise, I would appreciate their clarification.
> >
> > 4. > difference between titans (1) model-level, (2) context-level ... no code making experimental comparison difficult
> >
> > I appreciate the authors clarification of the differences along with the supporting table. I understand comparison can be difficult given that these approaches did not release code. However, if the claims are that GradMem is model level and context level I would expect some layer level + token-level memory ablations in the spirit of Titans/ATLAS to validate this design choice.
> >
> > 5. Furthermore, layer and token level updates do not require backpropagation through the whole model at test time. Eg: LaCT (ICLR 2026 https://arxiv.org/abs/2505.23884, code open sourced). These works report throughput much higher than Mamba, while the throughput gains of GradMem over Mamba reported during the rebuttal are more modest.
> >
> > I appreciate the motivation and the ideas presented in this work, but overall I think this work misses rigorous comparisons would what make the claims of this work stronger.

---

> > > ### Author Response · Authors · 2026-04-05
> > >
> > > Q1.
> > > Yes, ARMT and Mamba are strong baselines. On KV retrieval, which directly tests memory capacity, GradMem is stronger despite using a much smaller state: on KV-retrieval 1024 floats (8 memory tokens) vs about 41k for Mamba, on NLP tasks Mamba state is about 737k vs 6k-24k for GradMem.
> > >
> > > When we explicitly control for state size, GradMem substantially outperforms Mamba even when Mamba is allowed a larger state. In additional experiments, we matched Mamba state size more closely to GradMem's. GradMem remains much stronger even when Mamba has up to 4x larger state, but when state size is matched Mamba mostly fails. We will add [results table](https://postimg.cc/VSXXQc5M) to the revision.
> > >
> > > Based on the current evidence, we see GradMem as preferable when the goal is to compress very dense information into a very small explicit memory/state.
> > >
> > > Q2.
> > > Yes, we used the same training data that we used for training all models on KV retrieval.
> > >
> > > Q3.
> > > We agree that K is formally a hyperparameter. But being a hyperparameter does not make reporting original results for K > 1 insignificant.
> > >
> > > In GradMem, K controls iterative refinement of the memory state by minimizing the WRITE loss. Increasing K has a non-trivial effect. It gives clear gains, while repeated forward-only passes in RMT do not. Figure 4(a) also shows that at 64 KV-pairs, training with K_train=1 is not enough to fully benefit from larger K_eval, so multi-step WRITE has to be learned.
> > >
> > > Thus, our point is not that other methods cannot use multiple steps in principle. To our knowledge, prior related work does not analyze this effect, while we do. We believe this improves understanding of iterative memory refinement.
> > >
> > > Q4.
> > > For layer-level memory, we ran an additional experiment. We replaced input-level memory tokens with test-time trainable per-layer memory implemented as KV-cache using the same model-level reconstruction objective. On KV retrieval with 4-layer, 128-d models, we compare:
> > > (1) input-level memory with 8 mem tokens (1024 params),
> > > (2) size-matched per-layer memory (1024 params), and
> > > (3) larger per-layer KV-cache with the same cache size as for 8 tokens (8192 params, 8x larger).
> > >
> > > Input-level memory is more parameter-efficient: at matched size, it consistently outperforms per-layer memory, especially on harder settings. At 32 KV-pairs, input-level memory reaches about 87/98EM for K=1/2 vs to 32/66 for the size-matched per-layer memory. If the per-layer state is 8x larger, it becomes strongest, as expected given its much larger capacity. [Figure](https://postimg.cc/sMh71Xhb).
> > >
> > > To compare with layer-/token-level updates, we already added TTT-linear in the rebuttal, and we now also add LaCT on KV-retrieval task:
> > >
> > > |Method|8|16|32|64|96|
> > > |-|-:|-:|-:|-:|-:|
> > > |TTT-linear|97.2±0.9|96.9±0.7|82.1±3.4|38.6±14.4|11.9±10.8|
> > > |LaCT|95.7±6.8|99.9±0.1|99.7±0.4|62.5±25.2|46.9±24.3|
> > > |LaCT (top 1)|99.9|99.9|99.9|91.6|74.9|
> > > |GradMem (K=1)|99.7±0.0|96.3±0.9|86.9±0.5|58.6±0.7|32.6±0.1|
> > > |GradMem (K=5)|100.0±0.0|100.0±0.0|99.9±0.1|99.1±0.3|88.4±2.3|
> > >
> > > LaCT is strong but high-variance, so we report both 3-run average and top-1. Its memory state size is about 49k vs 1024 for GradMem. The main takeaway is that GradMem with more WRITE steps remains stronger overall with a much smaller memory state.
> > >
> > > We chose input-level memory because it is the simplest architecture-preserving design. It does not require modifying the internal layers of the base model, adding per-layer memory modules/objectives, or training a new architecture from scratch. It lets us use pretrained models as is and we can directly use model-level signal for optimization without the need to use local layer-level signals. To our knowledge, prior TTT/neural-memory works do not optimize a single model-level memory state with a model-level reconstruction objective by backpropagating through the whole model at test time.
> > >
> > > We do not study GradMem token-level updates because, under our model-level reconstruction objective, they would require sequential backprop-based WRITE updates for each token, which is computationally prohibitive. Our design instead intentionally treats WRITE as a single context-level operation.
> > >
> > > Q5.
> > > We added direct latency measurements for LaCT and TTT-MLP/linear for ~1B models using official fast implementations: [Figure](https://postimg.cc/bS8sFbHQ).
> > >
> > > For a single READ after WRITE, LaCT/TTT-MLP/linear are generally faster than GradMem. However, GradMem amortizes better when the same context is reused across multiple queries. At 4096 tokens, GradMem (1B) is more efficient than LaCT (760M) and TTT-Linear (1.3B) after 5 and 10 READs, respectively, and has a faster WRITE phase than TTT-MLP (1.3B).
> > >
> > > LaCT/TTT-MLP/linear perform many token/chunk-level updates during context processing, while GradMem performs a single model-level optimization of memory for the whole context.
> > >
> > > We also note that latency plots in the rebuttal for GradMem and Mamba are on a log scale: GradMem becomes about 1.6x faster than Mamba.

---

### Official Review · Reviewer_ryn1 · 2026-03-11

**Soundness:** 3
**Presentation:** 3
**Significance:** 3
**Originality:** 3
**Overall Recommendation:** 4
**Confidence:** 4

**Summary:**

The authors propose GradMem, which introduces a compressive memory mechanism that writes long contexts into a compact set of memory tokens using test-time gradient descent while keeping model weights frozen. By optimizing a self-supervised reconstruction loss, GradMem outperforms forward-only methods in memory capacity and retrieval accuracy, demonstrating strong transferability to natural language tasks.

**Compliance With Llm Reviewing Policy:**

Affirmed.

**Key Questions For Authors:**

see Weakness

**Limitations:**

yes

**Strengths And Weaknesses:**

**Strength:**

1. The authors introduce the concept of Test-time Training (TTT) into memory updates for long-context compression. Furthermore, they show that performance can be enhanced by scaling test-time computational overhead.  These are novel to me.
2. To improve the efficiency of the proposed GradMem, the authors implemented a custom double-backward pass that is both more memory-efficient and faster than a naive eager implementation.
3. The overall presentation and logical flow of the paper are clear, making it highly accessible and reader-friendly.

**Weakness:**

1. My main concern is that while the paper demonstrates the potential of the proposed GradMem method on synthetic and selected natural language tasks, the scale of these experiments is relatively limited (with models at 160M and input lengths up to 256 tokens). The effectiveness of the approach in genuine long-context scenarios remains to be further validated.

2. The proposed method's performance seems somewhat unstable.

   - For instance, in Table 1, while a performance drop is expected as $N$ increases, it is unclear why GradMem (and RMT) exhibits a sharp decline even at relatively small values of $N$;

   - Furthermore, the results in Table 2 suggest that increasing the number of gradient steps $K$ or the memory size $m$ does not always lead to consistent improvements and occasionally introduces instability. Such unpredictable behavior might hinder the broader application and adoption of the model.

3. I am curious about the memory capacity comparisons across different methods. While the authors ensured that RMT and GradMem use the same number of memory vectors, there is a lack of clarity regarding the memory capacity of Mamba (its state)  and ARMT (associative matrices). Adding a detailed comparison (or explanation) of this aspect in the main text would be beneficial in demonstrating the actual effectiveness of the proposed method.

4. Could the authors elaborate more on what the meta-learned initialization $M_0$ (and w/o meta-learning) refers to, and why "w/o meta-learning" is called a "first-order approximation"?

5. On page 4, line 195, the authors mention using "separate prediction heads for the WRITE and READ phases." As for pretrained models, I am somewhat confused about whether these "prediction heads" refer to the original pretrained head within $f_\\theta$ . My understanding is that the authors may have used the original pretrained weights as initializations (or started from scratch) to fine-tune separate heads for the WRITE and READ phases respectively, and then kept these two sets of weights frozen during inference (test-time).

---

> ### Author Rebuttal · Authors · 2026-03-31
>
> Thank you for the positive assessment of the idea, the implementation effort, and the clarity of the paper.
>
> ### On scale, larger models, and longer contexts
>
> We agree that original experiments were limited in scale. At the same time, we now added text memorization results on Llama-3.2-1B / 3B and latency measurements up to 8B, showing evidence that GradMem is not limited to the 100M setting.
>
> In particular, GradMem can write and recover text with pretrained 1B-3B LLMs even with only 1-2 memory vectors and 1-2 WRITE steps, reaching 99% acc. on 32-token texts (please see our response to reviewer oBek “On scalability to larger models”). We also benchmark long-context on Llama-3-8B at 4k / 8k / 16k tokens. Here, GradMem becomes more efficient in total WRITE+READ latency after approximately 27 / 43 / 61 repeated READs from the same context. For full latency details, please see our response to reviewer oBek (“On latency”).
>
> So, while full large-scale validation remains future work, we now provide concrete evidence that GradMem can be meaningfully scaled in both model size and context length.
>
> Results for text memorization: (https://postimg.cc/G8TphrbG). Latency results for Llama-3.2-1B, Llama-3-8B: (https://postimg.cc/3d2QrNmj) and GPT-2/Mamba: (https://postimg.cc/FdkKsmhL).
>
> ### On instability, sharp drops, and task dependence of larger K
>
> In Table 1, RMT performance degrades sharply as the KV context grows, whereas GradMem degrades more gradually, especially with additional WRITE steps (x5). For GradMem, the remaining drop at larger KV sizes is better interpreted as a capacity bottleneck (compressing more information into a fixed-size memory with limited WRITE compute) rather than as optimization instability. This is consistent with our additional sweep over memory size and K (see reviewer y4zh, “On capacity scaling with memory size…”), where increasing K makes a fixed memory budget more effective but does not remove the bottleneck when memory is too small.
>
> For the NLP tasks in Table 2, the picture is mixed. After completing the previously missing runs, we see no uniform improvement from increasing K across all tasks. So, the previously missing LM run with K=2 gives validation cross-entropy 2.91±0.01, i.e. only a small change relative to K=1, whereas the clearest positive effect of larger K is on SQuAD. So our interpretation is not that larger K systematically introduces instability, but rather that its benefit is task-dependent. We will clarify this in the revision.
>
> ### On fairness of memory-capacity comparisons
>
> Thank you, we agree this should be clearer. The most controlled comparison in our paper is GradMem vs RMT, because both use the same base architecture and the same model-level memory-token interface, and differ only in the WRITE rule. This is the clean comparison that supports our main claim that gradient-based WRITE is stronger than forward-only WRITE under matched memory/state.
>
> By contrast, Mamba and ARMT use qualitatively different memory mechanisms, so their states are not directly matched to a single input-level memory state. For our 4-layer, hidden size 128 setting, their state sizes are also much larger (Mamba: 40960, ARMT: 198144, vs GradMem/RMT: 1024. Thus Mamba and ARMT are strong reference baselines, but have no perfectly matched capacity. We will make this explicit in the main text.
>
> Please, note that GradMem outperforms Mamba and ARMT on SQuAD and show strong results on other NLP tasks, with an order or two orders of magnitude smaller memory state.
>
> ### On M0 and the “w/o meta-learning” ablation
>
> We agree this was not stated precisely enough. M0 is a shared starting memory state learned in pre-training  which is updated by K WRITEs in test time.
>
> We also agree that “w/o meta-learning” is better described as “w/o full meta-learning (1st-order approximation)”. This ablation still performs test-time updates and still gives a first-order signal to M0, but lacks the full second-order MAML signal through WRITE. The poor scores show that this first-order approximation is not sufficient to learn a strong WRITE rule. For fuller distinction, please see reviewer y4zh (“On method definitions and ablations”).
>
> ### On separate WRITE and READ heads
>
> Yes, your understanding is essentially correct, and we will make this clearer in the paper. For pretrained models, the READ phase uses the original pretrained prediction head inside `f_theta` (standard LM head). When separate WRITE/READ heads are enabled, the WRITE head is initialized from that same head and then finetuned for reconstruction. At test time, both heads and the rest of the model are frozen; only memory states are updated.

---

> > ### Author Rebuttal · Reviewer_ryn1 · 2026-04-04
> >
> > Thank you for the authors' response. I will maintain my score.

---

### Official Review · Reviewer_y4zh · 2026-03-12

**Soundness:** 3
**Presentation:** 4
**Significance:** 3
**Originality:** 3
**Overall Recommendation:** 5
**Confidence:** 4

**Summary:**

This paper studies the use of soft tokens for test-time context compression. It is motivated by the classic Recurrent Memory Transformer and incorporates ideas from test-time training.

**Compliance With Llm Reviewing Policy:**

Affirmed.

**Final Justification:**

Thanks for the detailed rebuttal. It addressed most of my concerns and I'll increase my score.

**Key Questions For Authors:**

See above

**Limitations:**

yes

**Strengths And Weaknesses:**

**Strengths**

1. The writing is very clear, intuitive, and easy to follow.
2. The paper investigates an interesting problem and provides comprehensive and solid experimental results.

**Weaknesses**

1. Could the authors clarify the terms *“forward-only”* and *“meta-learning”*? My understanding is that forward-only RMT does not include the test-time training step for memory writing, while the remaining components are the same.

   * If so, in Table 1 the results for x1, W/O TTT should be the same as FORWARD-ONLY WRITE (RMT) x1?
   * Also, the large performance drop for x1, W/O META-LEARNING might be due to the model weights being frozen. Is this the correct interpretation?

2. It would be interesting to stress-test the scaling behavior with respect to memory size, input length, and the number of gradient descent steps (K), while maintaining acceptable exact-match performance. This could help quantify how much contextual information (i.e., how many bits) a fixed number of memory tokens can reliably store.

3. Since GradMem is expected to be most useful when (|C| >> |M|), it would strengthen the paper to include experiments with longer contexts (e.g., >1k tokens). Currently there seems to be a discrepancy between the evaluation setting (very short contexts) and the theoretical computational benefits of GradMem discussed in Appendix D.

**Minor**

1. In Table 1, why are some entries marked with “–” for the 4–8 KV-pair setting? Also, why does RMT with more memory updates show inconsistent results? Some additional intuition would be helpful.

2. In Table 2, what is the LM cross-entropy performance of GradMem (GPT-2) when increasing (K)?

---

> ### Author Rebuttal · Authors · 2026-03-31
>
> Thank you for the careful reading and constructive questions. We especially appreciate that you found the paper clear and intuitive, and that you viewed the problem and experiments as interesting and solid.
>
> ### On method definitions and ablations:
>
> Yes, your understanding of “forward-only RMT” is correct: it uses the same memory tokens, but writes them only by forward computation, with no per-example gradient descent at test time. So “x1, w/o meta-learning” is not the same as “FORWARD-ONLY WRITE (RMT) x1”. That ablation still performs one gradient-based WRITE step at test time, whereas RMT uses a purely forward rule.
>
> We also agree that “w/o meta-learning” is imprecise. A better name is “w/o full meta-learning (1st-order approximation)”. The ablation still updates memory at test time, and training still provides a first-order signal to the shared initialization M0. What is missing is the full second-order signal through the WRITE updates.
>
> The large drop is not because weights are frozen as weights are frozen at test time in both GradMem and this variant. The key difference is whether training uses the full second-order meta-learning signal. The poor scores suggest that, in our setting, a first-order approximation is not enough and the full second-order signal is important for learning a strong WRITE rule.
>
> ### On capacity scaling with memory size and WRITE steps:
>
> Thank you for suggesting this analysis. We ran additional experiments sweeping memory size m=1,2,4,8 and WRITE steps K=1,2 on associative retrieval with 8 and 16 KV-pairs. Increasing K consistently improves Exact Match at fixed memory, especially in the low-memory regime, but larger contexts still require larger memory. Thus, for 8 KV-pairs, K=2 reaches near-perfect performance already at m=2, whereas for 16 KV-pairs near-perfect performance requires at least m=4. So, capacity scales roughly linearly here. When memory is very small (m=1), extra WRITE steps help substantially but do not remove the bottleneck. While this does not give an estimate in bits, it provides an empirical estimate of how storable context scales with memory size and WRITE compute. We will include the full figure in the revision: (https://postimg.cc/zLBB6K6X).
>
> ### On longer contexts and the amortized repeated-READ regime:
>
> GradMem is most useful not only when |C| >> |M|, but also when the same written context is reused across multiple READs.
>
> To complement the analytical break-even discussion in Appendix D, we measured long-context latency on GradMem built on Llama-3-8B at 4k / 8k / 16k tokens. GradMem WRITE+READ takes 1870 / 4860 / 14400 ms, compared to 364 / 747 / 1620 ms for Llama-3-8B and becomes more efficient after about 27 / 43 / 61 repeated READs, respectively, on an A100 with batch size 1.
>
> The reason is that GradMem’s READ cost depends on a fixed number of memory tokens, whereas Transformer KV-cache decoding grows with context length. Llama-3-8B takes about 49.1 ms to decode from an 8k-token cache and 76.6 ms from a 16k-token cache, while GradMem decodes from compressed memory in about 23 ms across all tested lengths. For full latency details, please see our response to reviewer oBek (“On latency”). Latency for Llama-3-8B and Llama-3.2-1B are here: (https://postimg.cc/3d2QrNmj).
>
> More broadly, long-context scaling remains a main challenge for meta-learning in Transformers despite the acceleration methods in App. C, as noted in our discussion. Cheaper approximations and zero-order alternatives to gradient-based WRITE are promising future directions.
>
> ### On missing Table 1 entries and LM with larger K:
>
> We have now added several of the previously missing runs. Table 1 now includes 4- and 8-pair settings for GradMem and RMT results up to 96 KV pairs. GradMem reaches 100% EM at 4 pairs and 99.7% at 8 pairs, and additional test-time gradient steps recover 100% on both. We added RMT results for >32 KV-pairs, and RMT degrades sharply beyond 32 pairs, reaching only 19.3% at 64 pairs and 12.9% at 96 pairs. Updated Table 1 is here: (https://postimg.cc/d7x1YDcG).
>
> We have also now completed the previously missing language modeling run with K=2. The validation cross-entropy is 2.91±0.01, i.e., only a small change relative to K=1, whereas the clearest positive effect is on SQuAD, where larger K gives a much more noticeable improvement. Updated Table 2 values are here: (https://postimg.cc/DJ4zyF7Q).
>
> ### On repeated RMT passes:
>
> In repeated forward passes with RMT, memory is replaced rather than accumulated. As a result, the same fixed WRITE rule is reapplied without an explicit per-example error signal, so extra passes can overwrite or interfere with useful memory content instead of steadily correcting it. This is why extra forward passes give limited or inconsistent gains, whereas additional gradient-based WRITE steps in GradMem are much more reliable.

---

> > ### Author Rebuttal · Reviewer_y4zh · 2026-04-04
> >
> > Thanks for the detailed rebuttal. It addressed most of my concerns and I'll increase my score.

---

### Decision · Program_Chairs · 2026-04-30

**Decision:**

Accept (regular)

**Comment:**

This paper introduces GradMem, a smart way to compress long text contexts into just a few memory tokens using test-time gradient descent instead of relying on a massive, memory-heavy cache. The reviewers liked the core idea and the clear writing, but they were initially worried about whether it would scale to larger models and if running a backward pass during inference would be too slow in practice. However, the authors did a good job in the rebuttal by testing on much larger Llama models and proving that the method actually saves a lot of time when you need to ask multiple questions about the exact same text. While one reviewer still had some lingering concerns about how the speed compares to other brand-new, layer-level methods, the rest of the committee felt the added experiments proved the method's practical value. Given these points, my recommendation is acceptance.